



# Evaluating the impact of post-processing medium-range ensemble streamflow forecasts from the European Flood Awareness System

Gwyneth Matthews[1], Christopher Barnard[2], Hannah Cloke[1,2,3,4,5], Sarah L Dance[1,6], Toni Jurlina[2], Cinzia Mazzetti[2], and Christel Prudhomme[2,7,8]

[1]Department of Meteorology, University of Reading, Reading, United Kingdom
[2]European Centre for Medium-range Weather Forecasts, Reading, United Kingdom
[3]Department of Geography and Environmental Science, University of Reading, Reading, United Kingdom
[4]Department of Earth Sciences, Uppsala University, Uppsala, Sweden
[5]Centre of Natural Hazards and Disaster Science, CNDS, Uppsala, Sweden
[6]Department of Mathematics and Statistics, University of Reading, Reading, United Kingdom
[7]Department of Geography, University of Loughborough, Loughborough, United Kingdom
[8]UK Centre for Ecology and Hydrology, Wallingford, United Kingdom

**Correspondence:** Gwyneth Matthews (g.r.matthews@pgr.reading.ac.uk)

**Abstract.** Streamflow forecasts provide vital information to aid emergency response preparedness and disaster risk reduction. Medium-range forecasts are created by forcing a hydrological model with output from numerical weather prediction systems. Uncertainties are unavoidably introduced throughout the system and can reduce the skill of the streamflow forecasts. Post-processing is a method used to quantify and reduce the overall uncertainties in order to improve the usefulness of the forecasts.

The post-processing method that is used within the operational European Flood Awareness System is based on the Model Conditional Processor and the Ensemble Model Output Statistics method. Using 2-years of reforecasts with daily timesteps this method is evaluated for 522 stations across Europe. Post-processing was found to increase the skill of the forecasts at the majority of stations both in terms of the accuracy of the forecast median and the reliability of the forecast probability distribution. This improvement is seen at all lead-times (up to 15 days) but is largest at short lead-times. The greatest improvement

was seen in low-lying, large catchments with long response times, whereas for catchments at high elevation and with very short response times the forecasts often failed to capture the magnitude of peak flows. Additionally, the quality and length of the observational time-series used in the offline calibration of the method were found to be important. This evaluation of the post-processing method, and specifically the new information provided on characteristics that affect the performance of the method, will aid end-users to make more informed decisions. It also highlights the potential issues that may be encountered

when developing new post-processing methods.



# 1 Introduction

Preparedness for floods is greatly improved through the use of streamflow forecasts resulting in less damage and fewer fatalities (Field et al., 2012; Pappenberger et al., 2015a). The European Flood Awareness System (EFAS), part of the European Commis-
sion's Copernicus Emergency Management Service, supports local authorities by providing continental-scale medium-range streamflow forecasts up to 15 days ahead (Thielen et al., 2009; Smith et al., 2016). These streamflow forecasts are produced by driving a hydrological model with an ensemble of meteorological forecasts from multiple numerical weather prediction (NWP) systems including two NWP ensembles and two deterministic NWP forecasts (Smith et al., 2016). However, the streamflow forecasts are subject to uncertainties that decrease their skill and limit their usefulness for end-users (Roundy et al., 2019;
Thiboult et al., 2017; Pappenberger and Beven, 2006). These uncertainties are introduced throughout the system and are often categorised as *meteorological uncertainties* (or input uncertainties) which propagate to the streamflow forecasts from the NWP systems, and *hydrological uncertainties* which account for all other sources of uncertainty including those from the initial hydrological conditions and errors in the hydrological model (Krzysztofowicz, 1999). It should be noted that throughout the paper meteorological uncertainties refers to the uncertainty in the streamflow forecasts that is due to the meteorological forc-
ings and not the uncertainty in the meteorological forecasts themselves. These differ as the meteorological variables are usually aggregated by the catchment system (Pappenberger et al., 2011). According to Krzysztofowicz (1999) and Todini (2008), a reliable forecast will include the total predictive uncertainty which is the probability of a future event occurring conditioned on all the information available when the forecast is produced.

Several approaches have been developed to reduce hydrological forecast errors and account for the predictive uncertainty.
Improvements to the NWP systems used to force the hydrological model have been shown to reduce the uncertainty in the streamflow forecasts (Dance et al., 2019; Flack et al., 2019; Haiden et al., 2021b). Additionally, the use of ensemble NWP systems to represent the uncertainty due to the chaotic nature of the atmosphere is becoming increasingly common and the use of multiple NWP systems can account for model parameter and structural errors in the meteorological forecasts (Wu et al., 2020; Cloke and Pappenberger, 2009). Regardless of whether deterministic or ensemble NWP systems are used, pre-processing
of the meteorological input can reduce biases and uncertainties often present in the forecasts (Verkade et al., 2013; Crochemore et al., 2016; Gneiting, 2014). Data assimilation schemes can be used to improve accuracy in the initial hydrological conditions (e.g. Liu et al., 2012; Mason et al., 2020) and calibration of the hydrological model can reduce model parameter uncertainties (Kan et al., 2019). To represent the hydrological uncertainties using an ensemble, similarly to the meteorological uncertainties, would require creating an ensemble of initial hydrological conditions and using several sets of model parameters or potentially
using multiple hydrological models (Georgakakos et al., 2004; Klein et al., 2020). Operationally this is usually prohibited by computational and temporal constraints particularly if an ensemble of meteorological forcings is already included. An alternative, relatively quick and computationally inexpensive approach is to post-process the streamflow forecasts.

Post-processing the streamflow forecast allows all uncertainties to be accounted for. Over the past few decades several techniques have been proposed. These techniques can be split into two approaches: (1) methods accounting for the meteorological
and hydrological uncertainties separately and (2) lumped approaches which calculate the total combined uncertainty of the





forecast. One of the first examples of the former approach was the Bayesian forecasting system which was applied to deterministic forecasts and consists of the Hydrological Uncertainty Processor (HUP Krzysztofowicz, 1999; Krzysztofowicz and Kelly, 2000; Krzysztofowicz and Herr, 2001; Krzysztofowicz and Maranzano, 2004) and an Input Uncertainty Processor (IUP Krzysztofowicz, 1999). The development of the Bayesian Ensemble Uncertainty Processor (Reggiani et al., 2009), an exten-
sion of the HUP for application in ensemble prediction systems, attempts to remove the need for the IUP by assuming the meteorological ensemble fully represents the input uncertainty. However, as streamflow forecasts are often under-spread this assumption is not always appropriate. The Model Conditional Processor (MCP) first presented in Todini (2008) also uses a conditional distribution-based approach by defining the joint distribution between the model output and the observations using a multi-variate Gaussian distribution. The MCP has the capacity to determine the total combined uncertainty if the joint distri-
bution is defined between the observations and the forecasts of the operational system. To define this joint distribution a large set of historic forecasts is required which is not always available as operational systems are upgraded regularly. Therefore, often it is used to account for the hydrological uncertainty only (as it is in this paper, see Sect. 3). However, the method is attractive as it can be efficiently extended to allow for multivariate, multi-model, and ensemble forecasts (Coccia, 2011; Coccia and Todini, 2011; Todini, 2013; Todini et al., 2015).

Many regression-based methods have been developed to post-process streamflow forecasts because of their relatively simple structure (e.g. quantile regression (Weerts et al., 2011), indicator cokriging (Brown and Seo, 2010, 2013), and the General Linear Model Post-Processsor (Zhao et al., 2011)). The Ensemble Model Output Statistics (EMOS, Gneiting et al., 2005) method adjusts the mean and variance of an ensemble forecast using linear functions of the ensemble members and the ensemble spread respectively (Gneiting et al., 2005; Hemri et al., 2015a). This allows variations in ensemble spread to be used when estimat-
ing the predictive uncertainty. The strong autocorrelation in time observed in hydrological timeseries lends itself to the use of autoregressive error-models (e.g. Seo et al., 2006; Bogner and Kalas, 2008; Schaeybroeck and Vannitsem, 2011) although some of these methods do not account for uncertainty and instead try to correct errors in the trajectory of the forecast. These methods should therefore be used alongside a separate method which attempts to quantify the uncertainty. On the other hand, kernel-based (or "dressing") methods define a kernel to represent the uncertainty which is superimposed over the forecast or
over every member for an ensemble forecast (Pagano et al., 2013; Verkade et al., 2017; Boucher et al., 2015; Shrestha et al., 2011). Depending on the approach used to define the kernel, this technique can account for the hydrological uncertainties or the total uncertainty but often requires a bias-correction method to be applied to the forecast beforehand (Pagano et al., 2013).

    All the methods mentioned above, and many more that have not been mentioned (see Li et al., 2017, for a more comprehensive review), have been shown to be effective at improving the skill of forecasts in one or a few catchments. The Hydrological
Ensemble Prediction Experiment (HEPEX, Schaake et al., 2007) post-processing intercomparison experiment resulted in comparisons between the different techniques (van Andel et al., 2013; Brown et al., 2013) but still relatively few studies have evaluated the performance of post-processing methods across many different catchments. Some exceptions include studies comparing the performance of post-processing techniques for limited numbers of basins in the USA (Brown and Seo (2013), 9 basins; Ye et al. (2014), 12 basins; and Alizadeh et al. (2020), 139 basins), and recently, Siqueira et al. (2021) evaluated two
post-processing methods at 488 stations across South America. However, as post-processing is incorporated into more large-





scale, multi-catchment flood forecasting systems, such as the EFAS, there is a greater need to understand which characteristics can affect the post-processing. In this paper, the operational post-processing method of the EFAS is evaluated at 522 stations to investigate how the performance of the post-processing method varies across the domain.

The EFAS domain covers hundreds of catchments across several hydroclimatic regions with different catchment charac-
teristics. The raw forecasts (i.e. forecasts that have not undergone post-processing) have varying levels of skill across these catchments (Alfieri et al., 2014), and are regularly evaluated in order to identify possible areas of improvement and to allow end-users to understand the quality of the forecasts. At the locations of river gauge stations, where near real-time and historic river discharge observations are available, the raw forecasts are post-processed using a post-processing method which is motivated by the MCP and EMOS techniques. However, the post-processed forecasts do not currently undergo regular evalu-
ation. This study aims to assess the post-processing method used within the EFAS. Additionally, new information is provided about the effect that characteristics of the catchments and properties of the forecasting system have on the performance of the post-processing method. Specifically, the paper will address the following questions:

– Does the post-processing method provide improved forecasts?

– What affects the performance of the post-processing method?

The remainder of the paper is set out as follows. In Sect. 2 we briefly describe the EFAS system used to produce forecasts operationally. In Sect. 3 we introduce the post-processing method being evaluated and explain in detail how the post-processed forecasts are created. In Sect. 4, the evaluation strategy is described. This includes an explanation of the criteria used to select stations, details of the reforecasts used in this evaluation, and a description of the evaluation metrics considered. We separate the results section (Sect. 5) into two main sub-sections. In Sect. 5.1 we assess the affect of post-processing on different features of the forecast such as the forecast median and the timing of the peak. In Sect. 5.2 we investigate how the benefits of post-
processing vary due to different catchment characteristics such as response time and elevation. Finally, in Sect. 6 we state our conclusion that post-processing improves the skill of the streamflow forecasts for most catchments and highlight the main factors affecting the performance of the post-processing method.

## 2   European Flood Awareness System (EFAS)

The EFAS produces a range of streamflow forecast products for catchments across Europe. The focus of this paper is the evaluation of the post-processing method used operationally to create the product referred to as the 'real-time hydrograph' (see Fig. 4). In the current section, we describe the operational production of the (raw) EFAS medium-range ensemble forecasts that are inputs for the post-processing method described in Sect. 3. Operationally, the medium-range ensemble forecasts are the foundation of several forecast products and are generated twice daily at 00 UTC and 12 UTC with a maximum lead-time of 15 days (Smith et al., 2016). Version 4 of EFAS (operational October 2020) produces forecasts with a 6-hourly timestep and at a resolution of 5 km x 5 km on a Lambert Azimuthal Equal Area Projection (Mazzetti et al., 2021). It should be noted that





post-processing is currently only performed at daily timesteps and requires the raw forecasts described here to be temporally aggregated.

The EFAS medium-range forecasts are created by driving a hydrological model, LISFLOOD, with the output from multiple

NWP systems. For the forecast, the required meteorological forcings are the precipitation, temperature, and potential evaporation fields from the NWP systems (Smith et al., 2016; EFAS, 2020). Operationally, four meteorological forecast products are used to drive LISFLOOD: two deterministic forecasts and two ensemble forecasts. The European Centre for Medium-range Weather Forecasts (ECMWF) provides (1) a deterministic forecast with a spatial resolution of 9 km and a maximum lead-time of 10 days, and (2) a 51-member ensemble forecast with a spatial resolution of 18 km and a maximum lead-time of 15 days

(EFAS, 2020). Both ECMWF forecasts are created using the ECMWF Integrated Forecasting System (currently Cycle 47r2, Haiden et al., 2021a; Maass, 2021). The German Weather Service (DWD) also provides (3) a high-resolution deterministic forecast. This forecast has a maximum lead-time of 7 days. The first 3 days are created using the COnsortium for Small-scale MOdelling - EUrope (COSMO-EU) model which has a spatial resolution 6.5 km (Baldauf and Wetterdienst, 2014; Deutscher Wetterdienst, 2015a). Days 4 to 7 are created using the DWD's ICON global model which has a spatial resolution 13 km

(Zängl et al., 2015; Deutscher Wetterdienst, 2015b). The final forecast is (4) a 20-member ensemble forecast produced using the COnsortium for Small-scale Modelling's Limited-area Ensemble Prediction System (COSMO-LEPS, Montani et al., 2011; COSMO, 2020). The COSMO-LEPS forecast has a maximum lead-time of 5 days and a spatial resolution of 7 km but has a smaller domain than the other NWP systems (see Fig. 1 in Smith et al. (2016) and Fig.1 in Arnal et al. (2019)). The forecast fields are regridded to fit the EFAS domain and used as input for the LISFLOOD hydrological model (Smith et al., 2016).

The LISFLOOD hydrological model is a GIS-based spatially distributed gridded rainfall-runoff-routing model specifically designed to replicate the hydrological processes of large catchments (Van Der Knijff et al., 2010; De Roo et al., 2000). The parameters controlling the processes within each grid-box are determined in an offline calibration of LISFLOOD which has recently been performed for the release of EFAS 4 (Mazzetti et al., 2021). For each grid-box in the EFAS domain LISFLOOD calculates the average discharge at 6-hourly timesteps.

The hydrological initial conditions for the streamflow forecasts are determined by forcing LISFLOOD with meteorological observations to create a simulation henceforth referred to as the *water balance simulation*. The water balance simulation provides the starting point of the forecast in terms of water storage within the catchment and discharge in the river. The precipitation and temperature point-observations (collected by the Meteorological Data Collection Centre of the Copernicus Emergency Management Service) are interpolated using the SPHEREMAP algorithm to the 5 x 5 km EFAS domain (Smith

et al., 2016; Arnal et al., 2019). The water balance simulation may contain errors due meteorological observation errors and structural errors in the hydrological model. Therefore, the water balance simulation may not represent the true state of the physical system, introducing uncertainty to the streamflow forecast. Operationally, there is a time delay in receiving the meteorological observations of 18 hours for the 00 UTC cycle and 30h for the 12 UTC cycle. The deterministic meteorological forecasts are used to drive the LISFLOOD model for the time period between the last available meteorological observation and

the initial timestep of the forecast in a process called the 'fill-up'.





# 3 Post-processing method

This section describes the post-processing method evaluated. Post-processing is performed at stations for which near real-time and historic river discharge observations are available. The method is motivated by the Multi-Temporal Model Conditional Processor (MT-MCP, Coccia, 2011) and Ensemble Model Output Statistics (EMOS, Gneiting et al., 2005) which are used to

quantify the hydrological and meteorological uncertainties, respectively. The Kalman filter is then used to combine these uncertainties. Since these methods assume Gaussianity, the Normal Quantile Transform (NQT) is used to transform the discharge values from physical space to standard Normal space. As with many post-processing methods, an offline calibration is required to define a so-called *station model*. In Sect. 3.1 some notation is introduced. Details on the post-processing method are given in Sects. 3.2 to 3.4. Figure 1 outlines the structure of the method. A discussion of the input data is postponed until Sect. 4.2.

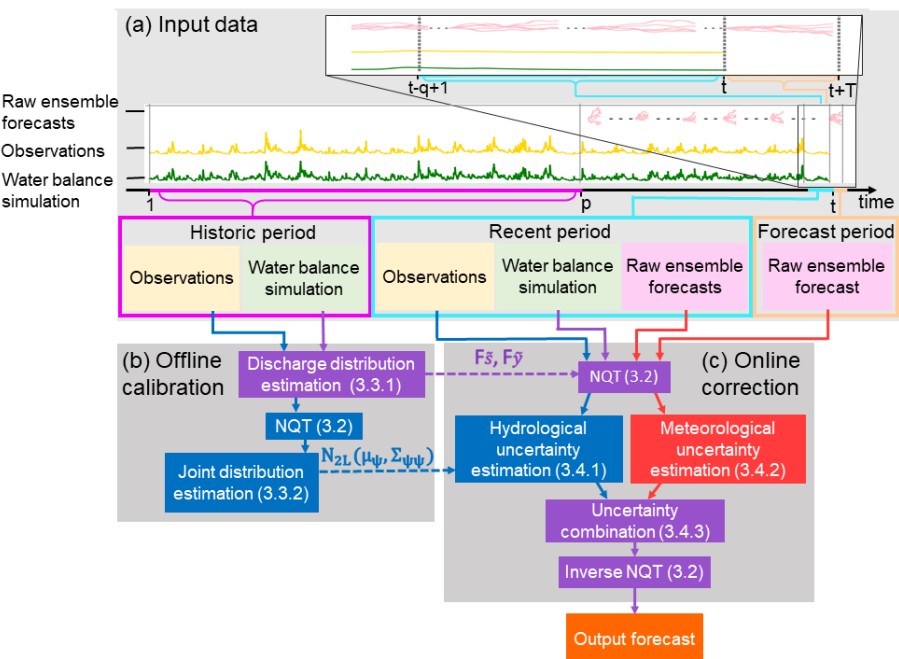

**Figure 1.** Flow chart describing the post-processing method at a station. (a) Input data are separated by time period (historic period: fuchsia, recent period: cyan, forecast period: peach) and by data type (observations: yellow, water balance simulations: green, raw ensemble forecasts: pink). The top time-series is a magnification of the bottom time-series for the period $t-q+1$ to $t+T$. The historic period has length $p$. For a forecast produced at time $t$, the recent period starts at time $t-q+1$ and the forecast period ends at time $t+T$. (b) Offline calibration steps. (c) Online correction steps. NQT is the Normal Quantile Transform. Blue and red arrows and boxes show the data and methods used to account for the hydrological uncertainty and the meteorological uncertainty, respectively. Data and methods used to account for both the hydrological and meteorological uncertainties are shown in purple. Dashed arrows show data stored in the station model such as the cumulative distribution functions of the water balance simulation and observations, denoted $F_{\tilde{s}}$, and $F_{\tilde{y}}$, respectively, and the joint distribution between the water balance simulation and observations, denoted $N_{2L}(\boldsymbol{\mu}_{\phi}, \boldsymbol{\Sigma}_{\phi\phi})$. Section numbers given in parentheses contain more details.





### 3.1 Notation

In this section notation and definitions used throughout the paper are introduced. The aim of post-processing is to correct the errors and account for the uncertainty that may be present in a forecast. As described in Sect. 2, the EFAS produces ensemble streamflow forecasts for the whole of Europe on a 5 km x 5 km grid with 6-hourly timesteps. However, post-processing is performed at daily timesteps and only at stations for which near real-time and historic river discharge observations are available. Therefore, the discharge values corresponding to the grid-boxes representing the locations of the stations are extracted and temporally aggregated to daily timesteps. This creates a separate streamflow forecast for each station and it is these single station forecasts that are henceforth referred to as the *raw forecasts*. The post-processing method evaluated in this paper is applied separately at each station creating a corresponding *post-processed forecast* for each raw forecast.

The input data shown in Fig. 1a is the input data required for the post-processing of a single raw forecast (i.e. for one station). As shown, the input data can be separated into three time periods. These time periods are henceforth referred to (from left to right in Fig. 1a) as the *historic period*, the *recent period*, and the *forecast period*. The length of the historic period, denoted $p$, varies between stations depending on the length of the historic observational record available. However, a minimum of 2 years of observations since 1991 is required for the offline calibration. For a forecast produced at time $t$, the recent period has $q$ timesteps and extends from time $t - q + 1$ to time $t$. The forecast period extends from time $t + 1$ to time $t + T$ for a forecast with a maximum lead-time of $T$ timesteps. The length of the recent period and the forecast period combined is $L = q + T$. For convenience, we introduce a *timestep notation* of the form $t_i : t_j$ to represent all timesteps between time $t_i$ and time $t_j$ i.e. $t_i : t_j$ means $t_i, t_i + 1, t_i + 2, \ldots t_j - 1, t_j$.

The raw ensemble forecast that is post-processed is the only data available in the forecast period. This forecast is produced at time $t$, has $M$ ensemble members, and a maximum lead-time of $T$ timesteps. The full ensemble forecast is represented by a matrix, denoted $\tilde{\mathbf{x}}_t(t + 1 : t + T) \in \mathbb{R}^{T \times M}$, where each column corresponds to an ensemble member and contains a vector of discharge values for each timestep in the forecast period. Here, the tilde notation indicates that the discharge values are in physical space, the subscript $t$ indicates the forecast production time, and the range of timesteps for which discharge values are available is shown using the timestep notation. The raw ensemble forecasts from the recent period are denoted using similar notation such that, for example, the forecast produced at $t - q + 1$ is denoted $\tilde{\mathbf{x}}_{t-q+1}(t - q + 2 : t - q + 1 + T) \in \mathbb{R}^{T \times M}$. All forecasts are from the same forecasting system and so all have $M$ ensemble members and maximum lead-times of $T$ timesteps.

The time-series of observations for a single station is denoted by the vector $\tilde{\mathbf{y}}$ where each element represents a daily discharge observation. The observations in the historic period are used in the offline calibration (see Fig. 1b and Sect. 3.3) and are denoted $\tilde{\mathbf{y}}(1 : p) \in \mathbb{R}^p$ where the timestep notation is used to show the range of timesteps for which observations are available. This vector is the same for all forecasts for this station as the station model is not updated between forecasts. The observations in the recent period (the $q$ timesteps up to the production time of the forecast) are used in the online correction (see Fig. 1c and Sect. 3.4) and are denoted $\tilde{\mathbf{y}}(t - q + 1 : t) \in \mathbb{R}^q$. Since $\tilde{\mathbf{y}}(t - q + 1 : t)$ is a function of $t$ the observations in this vector are different for each forecast production time.





Similarly, the time-series of the water balance simulation, denoted by the vector $\tilde{\mathbf{s}}$, is used in both the offline calibration and the online correction. Each element of the vector represents a daily water balance simulation value calculated by forcing

LISFLOOD with meteorological observations (see Sect. 2). The water balance simulation values from the historic period, $\tilde{\mathbf{s}}(1 : p)$, are selected to correspond to the timesteps of the $p$ observations from the same period. The water balance simulation values from the recent period are denoted $\tilde{\mathbf{s}}(t - q + 1 : t)$ and are dependent on the forecast production time, $t$.

### 3.2 Normal Quantile Transform (NQT)

The methods used in this post-processing method utilise the properties of the Gaussian distribution but discharge values usually

have highly skewed non-Gaussian distributions (Hemri, 2018). Therefore, the NQT is used to transform the discharge data to the standard Normal distribution which has a mean of zero and a variance of 1, denoted $N(0, 1)$. The NQT is applied separately to all input data (observed, simulated, and forecast) for a given station, therefore, it is defined here for any scalar discharge value $\tilde{\eta}$. The transformed discharge values are distinguished from the discharge values in physical space by the removal of the tilde notation (e.g. $\eta$).

The NQT defines a one-to-one map between the quantiles of the Cumulative Distribution Function (CDF) of the discharge distribution in physical space, $F_{\tilde{\eta}}(\tilde{\eta})$, and the CDF of the standard Normal distribution, $Q(\eta)$. The scalar function $F_{\tilde{\eta}}$ is dependent on whether $\tilde{\eta}$ represents a modelled discharge value (simulated or forecast) or an observed discharge value. The calculation of the discharge distributions and their subsequent CDFs are described in Sect. 3.3.1. The NQT transforms each scalar discharge value such that

$$\eta = Q^{-1}\left(F_{\tilde{\eta}}(\tilde{\eta})\right). \tag{1}$$

After the forecast values have been adjusted by the post-processing method, the inverse NQT,

$$\tilde{\eta} = F_{\tilde{\eta}}^{-1}\left(Q(\eta)\right), \tag{2}$$

is applied to transform the discharge values from the standard Normal space back to the physical space (see Fig. 1c).

### 3.3 Offline calibration

The offline calibration (see Fig. 1b) has two main aims: to determine the distributions of the observed, $\tilde{\mathbf{y}}$, and simulated, $\tilde{\mathbf{s}}$, discharge values at a station, and to define the joint distribution between the transformed observations, $\mathbf{y}$, and the transformed water balance simulation, $\mathbf{s}$. These distributions are then stored in the station model for use in the online post-processing step (shown by dashed lines in Fig. 1). The input data required for the offline calibration is an historic record of observations for the station, denoted by the vector $\tilde{\mathbf{y}}(1 : p) \in \mathbb{R}^p$, and, for the same period, an historic time-series of the water balance simulation

for the grid-box representing the location of the station, denoted by the vector $\tilde{\mathbf{s}}(1 : p) \in \mathbb{R}^p$, where the tilde notation signifies the variables are initially in physical space. The length of these vectors, $p$, is equal to the number of data points in the historic records and varies between stations. A minimum of 2 years historic data is required to guarantee that $p >> L$ (see Sect. 3.1).





### 3.3.1 Discharge distribution approximation

The NQT requires the Cumulative Distribution Function (CDF) of the observed and simulated discharge values in physical
space, denoted $F_{\tilde{y}}$ and $F_{\tilde{s}}$ respectively, to be defined. This section describes the approach used to estimate these functions.

First, the discharge density distributions are estimated using the observations, $\tilde{\mathbf{y}}(1:p) \in \mathbb{R}^p$, and the water balance simula-
tion values, $\tilde{\mathbf{s}}(1:p) \in \mathbb{R}^p$, from the historic period. These historic time-series are often only a few years long and therefore may
not represent the full discharge distribution due to the relative rarity of larger discharge values. To avoid the issues that short
time-series commonly cause in the inverse NQT (discussed in Bogner et al., 2012), rather than using the empirical distribution
as was done in the original MCP method (Todini, 2008), an approximation of the discharge distribution is determined using
a method similar to that presented in MacDonald et al. (2011). The approximation method applies kernel density estimation
to the bulk of the distribution (Węglarczyk, 2018) and fits a Generalised type II Pareto Distribution (GPD) to the upper tail
(Kleiber and Kotz, 2003) to create a composite distribution (see Fig. 2). The GPD is an extreme value distribution that is fully
defined by three parameters: the location parameter $a$, the scale parameter $b$, and the shape parameter $c$. Within this composite
distribution the location parameter also serves as the breakpoint which separates the kernel density and the GPD, and is shown
in Fig. 2. The time-series of discharge values, $\tilde{\boldsymbol{\eta}}(1:p) \in \mathbb{R}^p$, is used here to described the distribution approximation which is
implemented as follows:

1. All values in the time-series, $\tilde{\boldsymbol{\eta}}$, are sorted into descending order with $\tilde{\eta}_1$ denoting the largest value in the time-series, $\tilde{\eta}_2$
   denoting the second largest value and so on.

2. A Gaussian kernel is centered at each data point such that

$$K_i(x) = \frac{1}{\sigma_{\tilde{\eta}}\sqrt{2\pi}} e^{-(x-\tilde{\eta}_i)^2/2\sigma_{\tilde{\eta}}^2} \tag{3}$$

where $K_i$ is the kernel centered at $\tilde{\eta}_i$, and $\sigma_{\tilde{\eta}}$ is the Silverman's "rule of thumb" bandwidth (Silverman, 1984). The
bandwidth is calculated using the in-built R function *bw.nrd0* (R Core Team, 2019; Venables and Ripley, 2002) and all
values in the time-series, $\tilde{\boldsymbol{\eta}}$.

3. The kernel density is estimated using a leave-one-out approach such that the density at $\tilde{\eta}_j$ is

$$P(\tilde{\eta}_j) = \frac{1}{p-1} \sum_{i \neq j} K_i(\tilde{\eta}_j). \tag{4}$$

This makes sure the density is not over-fitted to any individual data point.

4. To guarantee data points in the tail, the largest 10 values are always assumed to be in the upper tail of the distribution
   (within the GPD) and the next 990 values (i.e. $\tilde{\eta}_{11}$ to $\tilde{\eta}_{1000}$) are each tried as the location parameter, $a$, of the GPD.

5. For each test value of $a$:

   i The scale parameter, $b$, is determined analytically by the constraints that the density must be consistent at the
   breakpoint regardless of which of the two distributions is used, and the integral of the density distribution function
   with respect to discharge must be equal to 1.





ii The shape parameter, $c$, is determined numerically by finding the maximum likelihood estimate within the limits
of $-1 \leq c \geq \frac{b}{\tilde{\eta}_1}$ (de Zea Bermudez and Kotz, 2010). The upper limit guarantees the upper bound of the distribution
is greater than the maximum value in the time-series, $\tilde{\eta}_1$, and the lower limit constrains the number of values
considered to reduce the computational time required. The concentrated likelihood method is used (Takeshi, 1985).

This produces 990 sets of parameters.

6. Of the sets of parameters found in step 5, maximum likelihood is used to choose the most likely set of parameters ($a_{ML}$, $b_{ML}$, $c_{ML}$) to define the GPD fitted to the upper tail of the distribution. The likelihood function for the full distribution
is the product of the likelihood function of the kernel density and the likelihood function of the GPD weighted by their
contribution to the total density (MacDonald et al., 2011).

The six steps outlined above are applied separately to both the simulated time-series, $\tilde{\mathbf{s}}(1:p)$, and the observed time-series, $\tilde{\mathbf{y}}(1:p)$. Figure 2 illustrates the approximation method for the simulated discharge distribution for a single station.

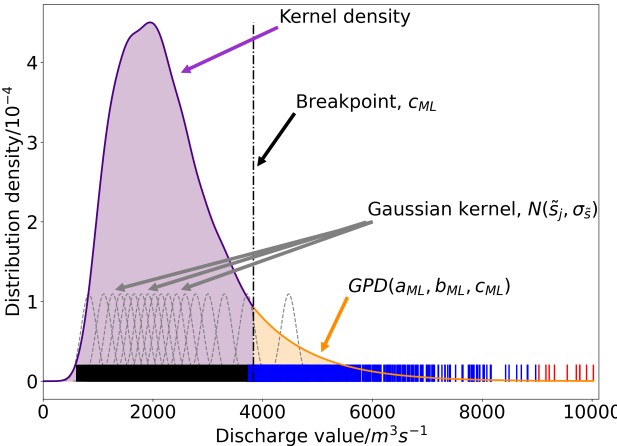

**Figure 2.** Schematic of the distribution approximation method. All data points are shown by the short solid lines. The largest 10 data points
are red (always in the upper tail), the next 990 largest data points are blue (tried as the location parameter), and the remaining data points
are black. Gaussian kernels (grey dashed lines) are used to calculate the kernel density (purple line). For clarity, only the kernels centered
at every $500^{th}$ data point are plotted. The upper tail is fitted with a Generalised type II Pareto distribution (orange line). The breakpoint
(dot-dashed black line) defines the separation between the two distributions. The integral of the density distribution function with respect to
discharge (the sum of the purple and orange shaded areas) equals 1.

Once the variables that define the discharge density distribution, namely $\sigma_{\tilde{\eta}}$, $a_{ML}$, $b_{ML}$, and $c_{ML}$, have been determined,
the Cumulative Distribution Function (CDF) can be calculated analytically for both the observed and simulated discharge
distributions. All input data (for both the online and offline parts of the method) must be transformed to the standard Normal
space using the NQT. However, it is too computationally expensive to calculate the analytical CDF for each data point. To
increase the computational efficiency of the NQT, both of the analytical CDFs are approximated as piecewise linear functions.





The approximated CDFs are linear between knots (boundary points between pieces of the piecewise function) and can therefore be defined by the knots and the CDF values at the knots. The approximated CDFs are calculated separately for the observed and simulated distributions as follows:

1. Each data point in the historic time-series, $\tilde{\boldsymbol{\eta}}(1:p)$ is considered a knot.

2. The CDF values at the mid-points between knots are approximated using linear interpolation and compared to the
analytical CDF values also calculated for the mid-points.

3. If the approximated CDF values at the midpoints are not within $1 \times 10^{-5}$ of the analytical CDF values then the mid-points are added as additional knots. Steps 2 and 3 are repeated until the approximated CDF values for all mid-points are within the allowed tolerance.

4. The knots and the analytical CDF values at the knots are saved in the station model.

Within the NQT (described in Sect. 3.2) the CDF for any discharge value, $F_{\tilde{\eta}}(\tilde{\eta})$, is approximated by linearly interpolating between knots. Ensuring that the CDF for any discharge value can be determined using linear interpolation makes the application of the NQT more efficient.

### 3.3.2   Joint distribution estimation

This section describes the calculation of the joint distribution used in the online hydrological uncertainty estimation (see Sect.
3.4.1). First, the discharge distributions defined in Sect. 3.3.1 are used within the NQT to transform the historic observations and water balance simulation to the standard Normal space (see Fig. 1b). This allows the joint distribution to be calculated as a multivariate Gaussian distribution. The joint distribution is defined between the observations and water balance simulation values at $L$ timesteps which, as noted in Sect. 3.1, is equal to the length of the recent period ($q$ timesteps) and forecast period ($T$ timesteps) combined. The $L$ timesteps are defined relative to a timestep $k$ such that the joint distribution is a $2L$-dimensional
distribution that describes the relationship between the observations, $\mathbf{y}(k-q+1:k+T)$, and the water balance simulation values, $\mathbf{s}(k-q+1:k+T)$. To ease notation we introduce the vector $\boldsymbol{\phi}(t_i:t_j)$, here defined generally for arbitrary timesteps, which includes the observed and simulated discharge values for all timesteps between timestep $t_i$ and timestep $t_j$, such that

$$\boldsymbol{\phi}(t_i:t_j) = \begin{pmatrix} \mathbf{y}(t_i:t_j) \\ \mathbf{s}(t_i:t_j) \end{pmatrix}. \tag{5}$$

Following on from Eq. (5), we define the vector $\boldsymbol{\psi} \in \mathbb{R}^{2L}$

$$\boldsymbol{\psi}(k-q+1:k+T) = \begin{pmatrix} \boldsymbol{\phi}(k-q+1:k) \\ \boldsymbol{\phi}(k+1:k+T) \end{pmatrix} = \begin{pmatrix} \mathbf{y}(k-q+1:k) \\ \mathbf{s}(k-q+1:k) \\ \mathbf{y}(k+1:k+T) \\ \mathbf{s}(k+1:k+T) \end{pmatrix} \in \mathbb{R}^{2L}. \tag{6}$$





The splitting of the observed and simulated variables into two distinct time periods i.e. timesteps of $k-q+1$ to $k$ and timesteps $k+1$ to $k+T$, is discussed below. The joint distribution can now be defined in terms of $\boldsymbol{\psi}(k-q+1:k+T)$.

The joint distribution is denoted $N_{2L}(\boldsymbol{\mu_\psi}(k-q+1:k+T),\boldsymbol{\Sigma_{\psi\psi}}(k-q+1:k+T,k-q+1:k+T))$ where the subscript $2L$ indicates its dimensions and the subscript $\boldsymbol{\psi}$ indicates that the distribution is for both observed and simulated variables for the two time periods shown in Eq. (6). The distribution is fully defined by its mean, $\boldsymbol{\mu_\psi}(k-q+1:k+T) \in \mathbb{R}^{2L}$, and covariance matrix, $\boldsymbol{\Sigma_{\psi\psi}}(k-q+1:k+T,k-q+1:k+T) \in \mathbb{R}^{2L\times 2L}$. Since both the observed and simulated historic timeseries have been transformed into the standard Normal space the mean discharge value is zero for both distributions and therefore the mean vector is defined as $\boldsymbol{\mu_\psi}(k-q+1:k+T) = \mathbf{0}$. The covariance matrix of the joint distribution is calculated using the historic observations and water balance simulation as

$$\boldsymbol{\Sigma_{\psi\psi}}(k-q+1:k+T,k-q+1:k+T) = \frac{1}{p-L}\sum_{k=q}^{p-T}\boldsymbol{\psi}(k-q+1:k+T)\boldsymbol{\psi}(k-q+1:k+T)^T \in \mathbb{R}^{2L\times 2L}, \qquad (7)$$

where $\boldsymbol{\psi}(k-q+1:k+T)$ is defined as in Eq. (6) for each timestep, $k$, in the historic period.

As mentioned, the joint distribution is used in the estimation of the hydrological uncertainty in the online part of the post-processing method (see Sect. 3.4.1). If the joint-distribution is defined such that $k$ is equal to the production time of a forecast then timesteps $k-q+1$ to $k$ correspond to the recent period and timesteps $k+1$ to $k+T$ correspond to the forecast period. Therefore, the joint distribution can be used to condition the unknown observations and water balance simulation values in the forecast period on the known observations and water balance simulation values from the recent period. Here, we introduce notation that is used to split the joint distribution into the variables corresponding to the recent period and the variables corresponding to the forecast period. Since the variables were split by timestep in the definition of $\boldsymbol{\psi}(k-q+1:k+T)$, the mean vector can also be split by timestep such that

$$\boldsymbol{\mu_\psi}(k-q+1:k+T) = \begin{pmatrix} \boldsymbol{\mu_\phi}(k-q+1:k) \\ \boldsymbol{\mu_\phi}(k+1:k+T) \end{pmatrix}, \qquad (8)$$

where $\boldsymbol{\mu_\phi}(k-q+1:k)$ represents the mean of the variables in the recent period for a forecast produced at time $k$ and $\boldsymbol{\mu_\phi}(k+1:k+T)$ represents the mean of the variables in the forecast period. The subscript $\phi$ indicates the distribution is for the observed and simulated variables for a single time period, following the structure shown in Eq. (5), rather than for both time periods as indicated by the subscript $\boldsymbol{\psi}$. The covariance matrix can be expressed as

$$\boldsymbol{\Sigma_{\psi\psi}}(k-q+1:k+T,k-q+1:k+T) = \begin{pmatrix} \boldsymbol{\Sigma_{\phi\phi}}(k-q+1:k,k-q+1:k) & \boldsymbol{\Sigma_{\phi\phi}}(k-q+1:k,k+1:k+T) \\ \boldsymbol{\Sigma_{\phi\phi}}(k+1:k+T,k-q+1:k) & \boldsymbol{\Sigma_{\phi\phi}}(k+1:k+T,k+1:k+T) \end{pmatrix} \qquad (9)$$

where $\boldsymbol{\Sigma_{\phi\phi}}(k-q+1:k,k-q+1:k)$ and $\boldsymbol{\Sigma_{\phi\phi}}(k+1:k+T,k+1:k+T)$ are the covariance matrices for variables in the recent and forecast periods, respectively, and $\boldsymbol{\Sigma_{\phi\phi}}(k-q+1:k,k+1:k+T)$ and $\boldsymbol{\Sigma_{\phi\phi}}(k+1:k+T,k-q+1:k)$ represent the cross-covariance matrices of variables in both time periods.





These submatrices can be further decomposed into the components referring to the observed and the simulated variables such that, for example,

$$
\boldsymbol{\Sigma}_{\phi\phi}(k+1:k+T, k+1:k+T) = \begin{pmatrix} \boldsymbol{\Sigma}_{\mathbf{yy}}(k+1:k+T, k+1:k+T) & \boldsymbol{\Sigma}_{\mathbf{ys}}(k+1:k+T, k+1:k+T) \\ \boldsymbol{\Sigma}_{\mathbf{sy}}(k+1:k+T, k+1:k+T) & \boldsymbol{\Sigma}_{\mathbf{ss}}(k+1:k+T, k+1:k+T) \end{pmatrix}, \tag{10}
$$

where the subscripts $\mathbf{y}$ and $\mathbf{s}$ indicate that the distribution refers to the observed and simulated variables, respectively (in contrast to the subscript $\phi$ which indicates both observed and simulated variables are included). The mean vector can also be split in this way such that

$$
\boldsymbol{\mu}_{\phi}(k+1:k+T) = \begin{pmatrix} \boldsymbol{\mu}_{\mathbf{y}}(k+1:k+T) \\ \boldsymbol{\mu}_{\mathbf{s}}(k+1:k+T) \end{pmatrix}. \tag{11}
$$

### 3.4  Online correction

This section describes the online correction part of the post-processing method (see Fig. 1c). The online correction quantifies and combines the hydrological and meteorological uncertainties for a specific forecast to produce the final probabilistic forecast. This forecast is produced at time $t$ and has a maximum lead-time of $T$ days, $\tilde{\mathbf{x}}_{t}(t+1:t+T) \in \mathbb{R}^{M \times T}$ (see Sect. 3.1 for a description of the notation). As shown in Fig. 1, the online correction requires the following input data from the recent period:

- observations for the station, $\tilde{\mathbf{y}}(t-q+1:t) \in \mathbb{R}^{q}$

- the water balance simulation for the grid-box containing the station's location, $\tilde{\mathbf{s}}(t-q+1:t) \in \mathbb{R}^{q}$

- a set of ensemble streamflow forecasts (from the same system as the forecast $\tilde{\mathbf{x}}_{t}$) for the grid-box containing the station's location, $\{\tilde{\mathbf{x}}_{t-q+1}, \tilde{\mathbf{x}}_{t-q+2}, \ldots, \tilde{\mathbf{x}}_{t-1}\}$.

All the input data is transformed to the standard Normal space using the NQT therefore the tilde notation is not used in the following sections. The CDFs determined in the offline calibration (see Sect. 3.3) and stored in the station model, $F_{\tilde{y}}$ and $F_{\tilde{s}}$, are used in the NQT to transform the variables as defined in Eq. (1). The observations are transformed using $F_{\tilde{y}}$, and the water balance simulation and forecasts are transformed using $F_{\tilde{s}}$. The following sections provide more detail on the methods used to account for the uncertainties and are performed within the standard Normal space. For simplicity, it is assumed that all data is available and there are no data latency issues such that the most recent observation available is $\tilde{y}(t)$ for the timestep when the forecast is produced. In practice, some observations from the recent period may not be available, and additionally the operational system does have data latency of approximately 1 day.

### 3.4.1  Hydrological uncertainties

The hydrological uncertainty is quantified using an MCP method which uses the discharge values from the recent period and the joint distribution, $N_{2L}(\boldsymbol{\mu}_{\boldsymbol{\psi}}, \boldsymbol{\Sigma}_{\boldsymbol{\psi\psi}})$, defined in the offline calibration (see Sect. 3.3.2). The joint distribution defines the





relationship between the observations and water balance simulation across $L$ timesteps where, as stated in Sect. 3.1, $L$ is equal to the length of the recent period ($q$ timesteps) and the forecast period ($T$ timesteps) combined. The hydrological uncertainty is estimated by conditioning the unknown observations and water balance simulation values in the forecast period on the known observed and simulated discharge values from the recent period using the joint distribution. First, the observations and water

balance simulations for the station from the recent period are combined into a single vector, $\boldsymbol{\phi}(t-q+1:t)$, as defined in Eq. (5).

In Sect. 3.3.2, the $L$ timesteps of the joint distribution were defined relative to a timestep $k$. Here, $k$ is set equal to the production time of the forecast, $t$, such that the timesteps from $t-q+1$ to $t$ correspond to the recent period and the timesteps from $t+1$ to $t+T$ correspond to the forecast period. Thus, the mean vector of the joint distribution can be expressed, as

discussed in Sect. 3.3.2, as

$$\boldsymbol{\mu}_{\psi}(t-q+1:t+T) = \begin{pmatrix} \boldsymbol{\mu}_{\phi}(t-q+1:t) \\ \boldsymbol{\mu}_{\phi}(t+1:t+T) \end{pmatrix} \tag{12}$$

where $\boldsymbol{\mu}_{\phi}(t-q+1:t)$ represents the mean of the variables (both observations and water balance simulation) in the recent period, for which we have known values, $\boldsymbol{\phi}(t-q+1:t)$, and $\boldsymbol{\mu}_{\phi}(t+1:t+T)$ represents the mean of the variables in the forecast period, which we are required to predict.

The sub-matrices of the covariance matrix of the joint distribution that were defined in Eq. (10) are also positioned relative to timestep $t$, such that,

$$\boldsymbol{\Sigma}_{\psi\psi}(t-q+1:t+T,t-q+1:t+T) = \begin{pmatrix} \boldsymbol{\Sigma}_{\phi\phi}(t-q+1:t,t-q+1:t) & \boldsymbol{\Sigma}_{\phi\phi}(t-q+1:t,t+1:t+T) \\ \boldsymbol{\Sigma}_{\phi\phi}(t+1:t+T,t-q+1:t) & \boldsymbol{\Sigma}_{\phi\phi}(t+1:t+T,t+1:t+T) \end{pmatrix}. \tag{13}$$

By positioning the joint distribution in this way, $\boldsymbol{\mu}_{\phi}(t+1:t+T) \in \mathbb{R}^{2T}$ and the sub-matrix $\boldsymbol{\Sigma}_{\phi\phi}(t+1:t+T,t+1:t+T) \in \mathbb{R}^{2T \times 2T}$ create a climatological forecast for the observations and water balance simulation in the standard Normal space. It is

this climatological forecast that is conditioned on the discharge values from the recent period.

The conditional distribution of the unknown discharge values in the forecast period conditioned on the known discharge values in the recent perion, denoted $N_{2T}(\widehat{\boldsymbol{\mu}}_{\phi}(t+1:t+T),\widehat{\boldsymbol{\Sigma}}_{\phi\phi}(t+1:t+T,t+1:t+T))$, is calculated using the properties of a multivariate Gaussian joint distribution (Dey and Rao, 2006) such that

$$\widehat{\boldsymbol{\mu}}_{\phi}(t+1:t+T) = \boldsymbol{\mu}_{\phi}(t+1:t+T)+$$

$$\boldsymbol{\Sigma}_{\phi\phi}(t+1:t+T,t-q+1:t)\boldsymbol{\Sigma}_{\phi\phi}(t-q+1:t,t-q+1:t)^{-1}\Big(\boldsymbol{\phi}(t-q+1:t)-\boldsymbol{\mu}_{\phi}(t-q+1:t)\Big) \tag{14}$$

and

$$\widehat{\boldsymbol{\Sigma}}_{\phi\phi}(t+1:t+T,t+1:t+T) = \boldsymbol{\Sigma}_{\phi\phi}(t+1:t+T,t+1:t+T)-$$

$$\boldsymbol{\Sigma}_{\phi\phi}(t+1:t+T,t-q+1:t)\boldsymbol{\Sigma}_{\phi\phi}^{-1}(t-q+1:t,t-q+1:t)\boldsymbol{\Sigma}_{\phi\phi}(t-q+1:t,t+1:t+T). \tag{15}$$





where the hat notation indicates it is conditioned on the discharge values from the recent period.

The resulting predicted distribution, $N_{2T}(\widehat{\boldsymbol{\mu}}_{\boldsymbol{\phi}}(t+1:t+T), \widehat{\boldsymbol{\Sigma}}_{\boldsymbol{\phi\phi}}(t+1:t+T, t+1:t+T))$ is referred to as the *hydrological uncertainty distribution* and can be partitioned into two $T$-dimensional forecasts; one for the water balance simulation and one for the unknown observations in the forecast period such that,


$$
\begin{bmatrix} \mathbf{y}(t+1:t+T) \\ \mathbf{s}(t+1:t+T) \end{bmatrix} \sim N_{2T}\left( \begin{bmatrix} \widehat{\boldsymbol{\mu}}_{\mathbf{y}}(t+1:t+T) \\ \widehat{\boldsymbol{\mu}}_{\mathbf{s}}(t+1:t+T) \end{bmatrix}, \begin{bmatrix} \widehat{\boldsymbol{\Sigma}}_{\mathbf{yy}}(t+1:t+T, t+1:t+T) & \widehat{\boldsymbol{\Sigma}}_{\mathbf{ys}}(t+1:t+T, t+1:t+T) \\ \widehat{\boldsymbol{\Sigma}}_{\mathbf{sy}}(t+1:t+T, t+1:t+T) & \widehat{\boldsymbol{\Sigma}}_{\mathbf{ss}}(t+1:t+T, t+1:t+T) \end{bmatrix} \right).
$$
(16)

The subscripts $\mathbf{y}$ and $\mathbf{s}$ indicate that the distribution refers to the observed and simulated variables, respectively. This notation was introduced in Sect. 3.3.2.

### 3.4.2 Meteorological uncertainty

This section describes the part of the online correction that estimates the meteorological uncertainty in the forecast of interest. As stated at the beginning of Sect. 3.4, the forecast of interest and the input data from the recent period are transformed into standard Normal space. The full transformed forecast, denoted by the forecast matrix $\mathbf{x}_t(t+1:t+T) \in \mathbb{R}^{T \times M}$ where each column represents an ensemble member (see Sect. 3.1), has ensemble mean $\overline{\mathbf{x}}_{\mathbf{t}}(t+1:t+T) \in \mathbb{R}^T$. The $i$-th component of $\overline{\mathbf{x}}_{\mathbf{t}}(t+1:t+T)$ represents the ensemble mean discharge at the $i$-th lead-time and is calculated as

$$
\overline{\mathbf{x}}_{\mathbf{t}}(t+1:t+T)[i] = \frac{1}{M} \sum_{m=1}^{M} \mathbf{x}_t(t+1:t+T)[i,m]
$$
(17)

The auto-covariance matrix of the forecast, $\boldsymbol{\Gamma}_{\boldsymbol{t}}(t+1:t+T, t+1:t+T) \in \mathbb{R}^{T \times T}$, is calculated such that the element corresponding to the $i$-th row and $j$-th column is given by

$$
\boldsymbol{\Gamma}_{\boldsymbol{t}}(t+1:t+T, t+1:t+T)[i,j] = \frac{1}{M-1} \sum_{m=1}^{M} (\mathbf{x}_t(t+1:t+T)[i,m] - \overline{\mathbf{x}}_{\mathbf{t}}(t+1:t+T)[i])(\mathbf{x}_t(t+1:t+T)[j,m] - \overline{\mathbf{x}}_{\mathbf{t}}(t+1:t+T)[j])^T.
$$
(18)

The uncertainty that propagates through from the meteorological forcings is partially captured by the spread of the ensemble 400 streamflow forecast. However, these forecasts are often under-spread particularly at shorter lead-times. The Ensemble Model Output Statistics method (EMOS, Gneiting et al., 2005) is used here to correct the spread only. Biases from the hydrological model are ignored in this section as the same hydrological model is used to create the water balance simulation and the forecasts. It is assumed that there is no bias in the meteorological forcings relative to the meteorological observations that are used to produce the water balance simulation (see Sect. 2) and that each ensemble member is equally likely. These assumptions 405 allow the value of the water balance simulation at any time $k$ to be expressed as

$$
s(k) = \overline{\mathbf{x}}_l(k) + \epsilon
$$
(19)





where $\overline{x}_i(k)$ is the ensemble mean for the timestep $k$ of a forecast produced at time $l$ (where $l+1 <= k <= l+T$), and $\epsilon$ is an unbiased Gaussian error. The value of the ensemble mean at timestep $k$, $\overline{x}_l(k)$, is therefore a random variable from the distribution $N(s(k), \sigma_\epsilon^2)$.

The variance of $\epsilon$, $\sigma_\epsilon^2$, should equal the expected value of the spread of the forecast, $E[\mathbf{\Gamma_t}]$. However, this is not always satisfied. To correct the spread, a set of forecasts from the recent period are used to estimate two spread correction parameters. The corrected covariance matrix, $\mathbf{\Gamma_t^c}(t+1:t+T, t+1:t+T) \in \mathbb{R}^{T \times T}$, is then calculated, using these spread correction parameters, such that

$$\mathbf{\Gamma_t^c}(t+1:t+T, t+1:t+T) = \zeta\left(\delta\mathbf{I} + \mathbf{\Gamma_t}(t+1:t+T, t+1:t+T)\right) \tag{20}$$

where $\mathbf{I}$ is the identity matrix, and $\zeta$ and $\delta$ are the scalar spread correction parameters to be determined.

The ensemble mean at each lead-time and the auto-covariance matrices are calculated for each of the forecasts from the recent period after they have been transformed to the standard Normal space (not including the forecast produced at time $t$ that is being corrected). Using the concentrated likelihood method (Takeshi, 1985) the spread correction parameters are defined as the maximum likelihoood estimates, $\zeta_{ML}$ and $\delta_{ML}$, for the likelihood function

$$L(\zeta, \delta | \{\mathbf{x_{t-q+1}}, \ldots, \mathbf{x_{t-1}}\}) = \prod_{k=t-q+1}^{t-1} \frac{1}{\sqrt{2\pi\zeta(\delta\mathbf{I} + \mathbf{\Gamma_k})}} \exp\left(-\frac{1}{2\zeta(\delta\mathbf{I} + \mathbf{\Gamma_k})}(\overline{\mathbf{x}}_k - \mathbf{s})^2\right) \tag{21}$$

where we have used a shorthand notation for clarity, such that $\overline{\mathbf{x}}_k = \overline{\mathbf{x}}_k(k+1:k+T)$, $\mathbf{\Gamma_k} = \mathbf{\Gamma_k}(k+1:k+T, k+1:k+T)$, and $\mathbf{s} = \mathbf{s}(k+1:k+T)$ as defined above.

The current forecast, $\mathbf{x_t}(t+1:t+T)$, is spread-corrected to account for the meteorological uncertainty by applying the parameters, $\zeta_{ML}$ and $\delta_{ML}$, as described in Eq. (20). This resultant distribution is referred to as the *meteorological uncertainty*
*distribution* and provides a prediction of the water balance simulation in the forecast period, such that

$$\mathbf{s}(t+1:t+T) \sim N(\overline{\mathbf{x}}_\mathbf{t}(t+1:t+T), \mathbf{\Gamma_t^c}(t+1:t+T, t+1:t+T)). \tag{22}$$

### 3.4.3    Combining uncertainties

The update step equations of the Kalman Filter (Kalman, 1960) are used to combine the hydrological and meteorological uncertainties to produce the final probabilistic forecast. The hydrological uncertainty distribution, defined in Eq. (16) and
denoted $N_{2T}(\widehat{\boldsymbol{\mu}}_{\boldsymbol{\phi}}(t+1:t+T), \widehat{\boldsymbol{\Sigma}}_{\boldsymbol{\phi\phi}}(t+1:t+T, t+1:t+T))$, is a predicted distribution for the water balance simulation and the observations during the forecast period. The meteorological uncertainty distribution, defined in Eq. (22) and denoted $N(\overline{\mathbf{x}}_\mathbf{t}(t+1:t+T), \mathbf{\Gamma_t^c}(t+1:t+T, t+1:t+T))$, is a predicted distribution for the water balance simulation in the forecast period. The predictions of the distribution of the water balance are compared within the Kalman filter. In order to extract the water balance simulation part of the hydrological uncertainty distribution we define the matrix "observation operator" $\mathbf{H}$ such
that

$$\widehat{\boldsymbol{\mu}}_\mathbf{s}(t+1:t+T) = \mathbf{H}\widehat{\boldsymbol{\mu}}_{\boldsymbol{\psi}}(t+1:t+T) = \mathbf{H}\begin{pmatrix} \widehat{\boldsymbol{\mu}}_\mathbf{y}(t+1:t+T) \\ \widehat{\boldsymbol{\mu}}_\mathbf{s}(t+1:t+T) \end{pmatrix} \in \mathbb{R}^T \tag{23}$$





where the subscripts $\mathbf{y}$ and $\mathbf{s}$ denote the observed and water balance simulation variables, respectively.

The update step of the Kalman filter is applied to produce a probabilistic forecast in the standard Normal space containing information about both the meteorological and hydrological uncertainties. The distribution of this forecast is denoted

$N_{2T}(\widehat{\boldsymbol{\mu}}_{\boldsymbol{\psi}}^{a}(t+1:t+T), \widehat{\boldsymbol{\Sigma}}_{\boldsymbol{\psi}\boldsymbol{\psi}}^{a}(t+1:t+T, t+1:t+T))$, where the superscript $a$ signifies the Kalman filter has been applied. The mean, $\widehat{\boldsymbol{\mu}}_{\boldsymbol{\psi}}^{a}(t+1:t+T))$, is calculated as

$$\widehat{\boldsymbol{\mu}}_{\boldsymbol{\psi}}^{a}(t+1:t+T) = \widehat{\boldsymbol{\mu}}_{\boldsymbol{\psi}}(t+1:t+T) + \mathbf{K}(\overline{\mathbf{x}}_{\mathbf{t}}(t+1:t+T) - \mathbf{H}\widehat{\boldsymbol{\mu}}_{\boldsymbol{\psi}}(t+1:t+T)) \tag{24}$$

where $\mathbf{K}$ is the Kalman gain matrix, defined as

$$\mathbf{K} = \widehat{\boldsymbol{\Sigma}}_{\boldsymbol{\psi}\boldsymbol{\psi}}(t+1:t+T, t+1:t+T)\mathbf{H}^{\mathbf{T}}(\mathbf{H}\widehat{\boldsymbol{\Sigma}}_{\boldsymbol{\psi}\boldsymbol{\psi}}(t+1:t+T, t+1:t+T)\mathbf{H}^{\mathbf{T}} + \boldsymbol{\Gamma}_{\mathbf{t}}^{\mathbf{c}}(t+1:t+T, t+1:t+T))^{-1}, \tag{25}$$

and $\mathbf{H}$ is the matrix observation operator defined above. The auto-covariance matrix is calculated as

$$\widehat{\boldsymbol{\Sigma}}_{\boldsymbol{\psi}\boldsymbol{\psi}}^{a}(t+1:t+T, t+1:t+T) = (\mathbf{I} - \mathbf{K}\mathbf{H})\widehat{\boldsymbol{\Sigma}}_{\boldsymbol{\psi}\boldsymbol{\psi}}(t+1:t+T, t+1:t+T) \tag{26}$$

where $\mathbf{I}$ is the identity matrix and all other symbols are as before. The distribution produced by combining these two sources of uncertainty, $N_{2T}(\widehat{\boldsymbol{\mu}}_{\boldsymbol{\psi}}^{a}(t+1:t+T), \widehat{\boldsymbol{\Sigma}}_{\boldsymbol{\psi}\boldsymbol{\psi}}^{a}(t+1:t+T, t+1:t+T))$, is for both the unknown observations and the water balance simulations variables in the forecast period. This distribution is partitioned into two $T$-dimensional forecasts which are

in the standard Normal space such that

$$\begin{bmatrix} \mathbf{y}(t+1:t+T) \\ \mathbf{s}(t+1:t+T) \end{bmatrix} \sim N_{2T}\left( \begin{bmatrix} \widehat{\boldsymbol{\mu}}_{\mathbf{y}}^{a}(t+1:t+T) \\ \widehat{\boldsymbol{\mu}}_{\mathbf{s}}^{a}(t+1:t+T) \end{bmatrix}, \begin{bmatrix} \widehat{\boldsymbol{\Sigma}}_{\mathbf{yy}}^{a}(t+1:t+T, t+1:t+T) & \widehat{\boldsymbol{\Sigma}}_{\mathbf{ys}}^{a}(t+1:t+T, t+1:t+T) \\ \widehat{\boldsymbol{\Sigma}}_{\mathbf{sy}}^{a}(t+1:t+T, t+1:t+T) & \widehat{\boldsymbol{\Sigma}}_{\mathbf{ss}}^{a}(t+1:t+T, t+1:t+T) \end{bmatrix} \right)$$
$$\tag{27}$$

where the subscripts $\mathbf{y}$ and $\mathbf{s}$ denote the observed and water balance simulation variables, respectively.

The $T$-dimensional distribution corresponding to the predicted distribution of the unknown observations in the forecast period, $N_T(\widehat{\boldsymbol{\mu}}_{\mathbf{y}}^{a}(t+1:t+T), \widehat{\boldsymbol{\Sigma}}_{\mathbf{yy}}^{a}(t+1:t+T, t+1:t+T))$, is transformed back into physical space using the inverse NQT, defined in Eq. (2), and the CDF of the observed discharge distribution, $F_{\tilde{y}}$. This forecast is then used to produce the 'real-time

hydrograph' (see Fig. 4 for an example of this forecast product).

## 4 Evaluation strategy

### 4.1 Station selection

To maintain similarity with the operational system, the station models used in this evaluation are those calibrated for use in the

operational post-processing. To avoid an unfair evaluation, station models must have been calibrated using observations from before the evaluation period. An evaluation period of approximately 2-years (from 1 January 2017 to 14 January 2019) was chosen to balance the length of the evaluation period with the number of stations evaluated. Of the 1200 stations post-processed





operationally, 610 stations have calibration time-series with no overlap with the evaluation period. Additionally, stations were required to have at least 95% of the daily observations for the evaluation period which reduced the number of stations to 525,

and a further three stations were removed after a final quality control inspection (see Sect. 4.2.2 for details of the observations and the quality control system used). Although all 522 stations are evaluated, specific stations are used to illustrate key results (see Sect. 5.2). These stations are labelled in Fig 3.

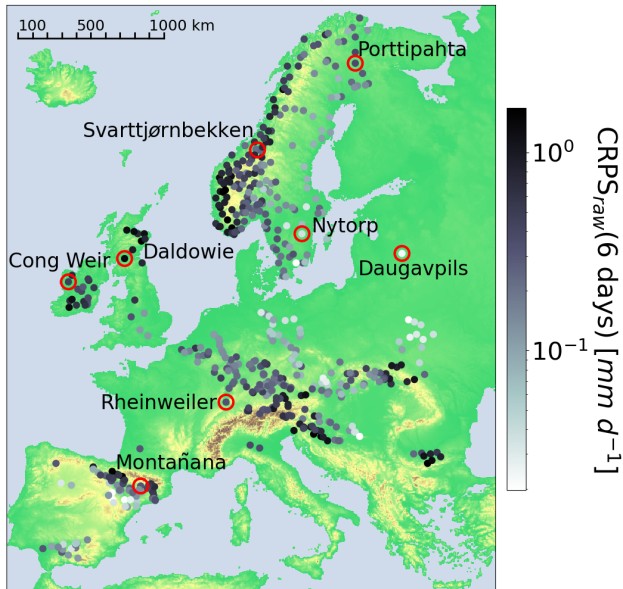

**Figure 3.** Map showing the locations of the 522 stations evaluated. The marker colour shows the Continuous Ranked Probability Score (see Sect. 4.3.4) for the raw forecast at a lead-time of 6 days on a log scale. Perfect score: CRPSS=0. Stations used as examples in Sect. 5 are labelled and highlighted by the red circles.

## 4.2 Data

### 4.2.1 Reforecasts

In this study, reforecasts were used to evaluate the post-processing method. Reforecasts are forecasts for past dates created using a forecasting system as close to the operational system as possible (Hamill et al., 2006; Harrigan et al., 2020). As the EFAS system has recently been updated, using reforecasts allowed a larger number of forecasts to be evaluated. The reforecasts used are a subset of the EFAS 4.0 reforecast dataset (Barnard et al., 2020). This dataset contains twice-weekly reforecasts for dates that correspond to each Monday and Thursday in 2019. For example, 3 January 2019 is a Thursday, so the dataset

contains reforecasts for 3 January for every year from 1999 to 2018. The chosen evaluation period (see Sect. 4.1) includes 208 reforecasts.



Using twice-weekly reforecasts, rather than daily, reduces the temporal correlations between forecasts and therefore limits the dependence of the results on the autocorrelation of the river discharge (Pappenberger et al., 2011). However, this means any single event cannot be included in the evaluation for all lead-times. For example, an event that occurs on a Saturday will not be included within the evaluation of the forecasts at a lead-time of 1 day which can only be a Tuesday or a Friday. Where necessary the evaluation metrics were combined over several lead-times (see Sects. 4.3.2 and 4.3.3). Additionally, fewer reforecasts were available to estimate the EMOS parameters in the meteorological uncertainty estimation (see Sect. 3.4.2). Whereas operationally there would be 40 forecasts corresponding to the previous $q = 40$ days, here only around 11 reforecasts were available within the recent period.

Each reforecast is an ensemble forecast of 11 members created by forcing LISFLOOD with reforecasts from the ECMWF ensemble NWP system. At every timestep the mean discharge value for the previous 6 hours is predicted for each 5 km x 5 km grid-box in the EFAS domain. Currently post-processing is only performed for daily timesteps. Therefore, the reforecasts were aggregated to daily timesteps with a maximum lead-time of $T = 15$ days. These ensemble reforecasts are henceforth referred to as the *raw forecasts*. The raw forecasts were used as input for the post-processing method to create the *post-processed forecasts*. Both sets of forecasts were evaluated as described in Sect. 4.3.

### 4.2.2 Observations

All discharge observations were provided by local and national authorities and collected by the Hydrological Data Collection Centre of the Copernicus Emergency Management Service and are the observations used operationally. The operational quality control process was applied to remove incorrect observations before they were used in this study (Arroyo and Montoya-Manzano, 2019; McMillan et al., 2012). Additionally, simple visual checks were performed to account for any computational errors introduced after the operational quality checks. Average daily discharge observations were used in three parts of the study. For each station, an historic time-series was used in the calibration of the station model (see Sect. 3.3). The length of the historic time-series, denoted $p$ in Sect. 3.1, varies in length between stations. However, a minimum of 2-years of observational data between 1 January 1990 and 1 January 2017 is required. For each reforecast, records of near real-time observations from the $q = 40$ days prior to the forecast time were used as the observations in the recent period (see Sect. 3.4.1). Observations from the evaluation period were used as the truth values in the evaluation (see Sect. 4.3).

### 4.2.3 Water balance simulation

The EFAS 4.0 simulation (Mazzetti et al., 2020) was used as the water balance simulation for dates between 1 January 1990 and 14 January 2019. As described in Sect. 2, the water balance simulation is created by driving LISFLOOD with gridded meteorological observations. This dataset provides simulations for the whole of the EFAS domain. The values for the grid-boxes representing the locations of the stations were extracted creating a simulated time-series for each station. These time-series were aggregated from 6-hourly timesteps to daily timesteps (00 UTC to 00 UTC) and were used in three ways in this study. The water balance values for dates corresponding to the available observations in the historic period were used to calibrate the station model (see Sect. 3.3). For dates within the recent period for each reforecast, the water balance values were





510  used in the post-processing (see Sect. 3.4.1). Finally, the water balance values corresponding to the 15 day lead-time of each reforecast was used to estimate the average meteorological error of each station (see Sect. 5.2.1).

## 4.3  Evaluation metrics

The evaluation of the post-processing method is performed by comparing the skill of the raw forecasts with the corresponding post-processed forecasts. Since the aim of the post-processing is to create a more accurate representation of the observation

515  probability distribution all metrics use observations as the "truth" values. As mentioned in Sect. 2, the output from the post-processing method evaluated here is expressed operationally in the 'real-time hydrograph' product, an example of which is shown in Fig. 4. Therefore, the evaluation will consider four main features of forecast hydrographs.

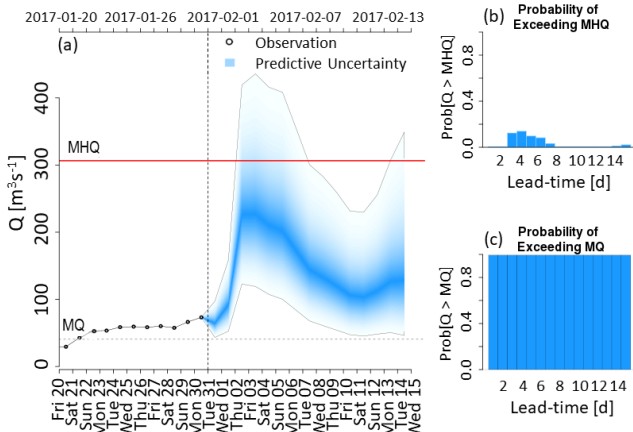

**Figure 4.** Example of the 'real-time hydrograph' product for the station in Brehy, Slovakia on 31 January 2017. (a) Probability distribution of the post-processed forecast. The darkest shade of blue indicates the forecast median (50th percentile) with each consecutive shade indicating a percentile difference such that the extent of the total predictive uncertainty is shown by the shaded region. Solid grey lines indicate the upper (99th percentile) and the lower (1st percentile) bounds of the forecast probability distribution. The red line shows the mean annual maximum (MHQ) threshold, and the dashed grey line shows the mean flow (MQ) threshold. Black circles represent observations positioned at the centre of the timestep over which they are calculated. (b) Bar chart showing the probability of the discharge exceeding the MHQ threshold at each lead-time. (c) Bar chart showing probability of the discharge exceeding the MQ threshold at each lead-time.

### 4.3.1  Forecast median

In the real-time hydrograph the darkest shade of blue indicates the forecast median making it the easiest and most obvious

520  single-valued summary of the full probabilistic forecast for end-users. The skill of the forecast median is evaluated using the modified Kling-Gupta Efficiency score ($KGE'$, Kling et al., 2012; Gupta et al., 2009). The forecast median is determined for the post-processed forecasts by extracting the 50th percentile of the probability distribution at each lead-time. For the raw forecasts the ensemble members are sorted by discharge value and the middle (i.e. 6th) member is chosen. This is done





separately for each lead-time so the overall trajectory may not follow any single member. The forecast median is denoted $x$ to
distinguish it from the full forecast $\mathbf{x_t}(t+1:t+T)$. The $KGE'$ is calculated as,

$$KGE' = 1 - \sqrt{(r-1)^2 + (\beta-1)^2 + (\gamma-1)^2} \tag{28}$$

with

$$\beta = \frac{\overline{x}}{\overline{y}}, \tag{29}$$

and

$$\gamma = \frac{\sigma_x/\overline{x}}{\sigma_y/\overline{y}}, \tag{30}$$

where $r$ is the Pearson's correlation coefficient, $\overline{x}$ and $\overline{y}$ are the mean values of the forecast median and the observations, re-
spectively, and $\sigma_x$ and $\sigma_y$ are their standard deviations. The correlation, $r$, measures the linear relationship between the forecast
median and the observations indicating the ability of the forecasts to describe the temporal fluctuations in the observations. The
bias ratio, $\beta$, indicates if the forecast consistently under or over-predicts the observations. The variability ratio, $\gamma$, measures
how well the forecast can capture the variability of the discharge magnitude. The $KGE'$ is calculated separately for each
lead-time. The $KGE'$ ranges from $-\infty$ to 1, $r$ ranges from -1 to 1, and both $\beta$ and $\gamma$ range from $-\infty$ to $\infty$. A perfect score
for the $KGE'$ and each of the components is 1.

### 4.3.2 Peak discharge

The timing of the peak discharge is an important variable of flood forecasts. The Peak-Time Error ($PTE$) is used to evaluate
the effect of post-processing on the ability of the forecast to predict the timing of the peak. The $PTE$ requires a single-valued
forecast trajectory. For the reasons stated in Sect. 4.3.1, the $PTE$ is calculated using the forecast median, $x$. Peaks are defined as
the maximum value in the forecast period for forecasts where the maximum observation exceeds the 90th percentile discharge
threshold of the station. This threshold is calculated using the full observational record for the station. The $PTE$ is calculated
as,

$$PTE = t_n^x - t_n^y \tag{31}$$

where $t_n^x$ is the timestep of the maximum of the forecast median for the $n$-th forecast and $t_n^y$ is the timestep of the maximum
observed value in the same forecast period. A perfect score is $PTE = 0$. A negative $PTE$ value indicates the peak is forecast
too early and a positive $PTE$ value indicates the peak is forecast too late. As the maximum lead-time is 15 days, the maximum
value of the $PTE$ is 14 days and the minimum value is -14 days.

### 4.3.3 Threshold exceedance

Two discharge thresholds are shown in the real-time hydrograph: the mean discharge (MQ) and the mean annual maximum
discharge (MHQ). Both thresholds are determined using the observations from the historic period. For the post-processed





forecasts, the probability of exceedance of the MQ threshold, $PoE(MQ)$, is calculated such that

$$PoE(MQ) = 1 - F_{\tilde{x}}(MQ) \tag{32}$$

where $F_{\tilde{x}}(MQ)$ is the value of the forecast CDF at the MQ threshold. The CDF is assumed to be linear between any two percentiles. The same method is applied for the MHQ threshold. For the ensemble forecast, each ensemble member above the threshold contributes one eleventh to the probability of the threshold being exceeded. The probability of the threshold being exceeded is calculated separately for each lead-time.

The Relative Operating Characteristic (ROC) score and ROC diagram (Mason and Graham, 1999) are used to evaluate the
potential usefulness of the forecasts with respect to these two thresholds. The ROC diagram shows the probability of detection vs the false alarm rate for alert trigger thresholds from 0.05 to 0.95 in increments of 0.1. The ROC score is the area below this curve with a ROC score of less than 0.5 indicating a forecast with less skill than a climatological forecast. As discharge values of above the MHQ threshold are rare, all stations are combined and lead-times are combined into 3 groups; 1-5 days, 6-10 days, and 11-15 days. Since the reforecasts are only produced on Monday and Thursdays, an event that occurs on a Saturday
can only be forecasted at lead-times of 2, 5, 9, and 12 days. Using 5-day groupings of lead-times guarantees that each group is evaluated against each event at least once but allows the usefulness of the forecasts to be compared at different lead-times. A perfect forecasting system would have a ROC score of 1.

Reliability diagrams are used to evaluate the reliability of the forecast in predicting the exceedance of the two thresholds. Reliability diagrams show the observed frequency vs the forecast probability for bins of width 0.1 from 0.05 to 0.95. A
perfectly reliable forecast would follow the one-to-one diagonal on a reliability diagram. The same combination of stations and lead-times is used as with the ROC diagrams.

### 4.3.4 Full probability distribution

A commonly used metric to evaluate overall performance of a probabilistic or ensemble forecast is the Continuous Ranked Probability Score (CRPS, Hersbach, 2000). The CRPS measures the difference between the CDF of the forecast and that of
the observation and is defined as

$$CRPS\left(F_{\tilde{x}}, y\right) = \int_{-\infty}^{\infty} \left(F_{\tilde{x}}\left(\tilde{\eta}\right) - \theta\left(\tilde{\eta} - y\right)\right)^2 d\tilde{\eta} \tag{33}$$

where $F_{\tilde{x}}$ represents the CDF of the forecast and $\theta(\tilde{\eta} - y)$ is the step function (Abramowitz and Stegun, 1972), defined such that

$$\theta(\tilde{\eta}) = \begin{cases} 0 & \tilde{\eta} < 0 \\ 1 & \tilde{\eta} \geq 0 \end{cases} \tag{34}$$

and represents the CDF of the observation, y. The post-processed forecasts are defined via their percentiles, therefore by assuming the CDF is linear between percentiles the CRPS can be calculated directly. The empirical CDF of the raw forecasts, defined via point statistics, is used and the CRPS is calculated using a computationally efficient form (Jordan et al., 2019, Equation 3). The CRPS ranges from a perfect score of 0 to $\infty$.





### 4.3.5 Comparison

For some of the metrics described in Sects. 4.3.1-4.3.4, the impact of post-processing is shown using the respective skill score, $SS$, with the raw forecast as the benchmark,

$$SS = \frac{S_{pp} - S_{raw}}{S_{perf} - S_{raw}} \tag{35}$$

where $S_{pp}$ and $S_{raw}$ are the scores for the post-processed forecast and the raw forecast, respectively, and $S_{perf}$ is the value of the score for a perfect forecast. The skill score gives the fraction of the gain in skill required for the raw forecast to become

a perfect forecast that is provided by the post-processing. A value $SS < 0$ means the forecast has been degraded by the post-processing, a value of $SS > 0$ indicates that the forecast has been improved by the post-processing, and a value of $SS = 1$ means that the post-processed forecast is perfect. Henceforth, the skill score for a metric is denoted by adding 'SS' to the metric name.

## 5 Results and Discussion

### 5.1 Performance of the post-processing method

This section focuses on the overall impact of post-processing at all 522 of the evaluated stations across the EFAS domain and aims to address the research question: *Does the post-processing method provide improved forecasts?*

### 5.1.1 Forecast median

The modified Kling-Gupta Efficiency Skill Score (KGESS) is used to evaluate the impact of post-processing on the forecast

median (see Sect. 4.3.1). Figure 5a shows the KGESS for all stations at every other lead-time such that each boxen plot (also known as letter-value plots, Hofmann et al., 2017) contains 522 values, one for each station. For each lead-time the central black line shows the median KGESS value. The inner box (the widest box) represents the interquartile range and contains 50% of the data points. Each subsequent layer of boxes splits the remaining data points in half such that the second layer of boxes are bounded by the 12.5th and 87.5th percentiles and contains 25% of the data points. The outliers represent a total of 2%

of the most extreme data points. The lower panels of Fig. 5 show the three components of the $KGE'$ (b: correlation, c: bias ratio, d: variability ratio) for lead-times of 3, 6, 10, and 15 days for all stations for both the raw forecasts (orange) and the post-processed forecasts (purple). The chosen lead-times are representative of the results.

Figure 5a shows that most stations have positive KGESS values at all lead-times indicating that post-processing increases the skill of the forecast median. However, the magnitude of this improvement decreases at longer lead-times. The median KGESS

value decreases from 0.75 at a lead-time of 1 day to 0.15 at a lead-time of 15 days with most of this reduction occurring in the first 7 days. Nonetheless, over 11% of stations have a KGESS greater than 0.5 at a lead-time of 15 days; most notably the station in Nytorp, Sweden has a KGESS above 0.98 at this lead-time. As discussed in Sect. 5.2.2, the large KGESS values at the Nytorp station are due to the correction of a large bias in the raw forecast. The proportion of stations for which post-processing





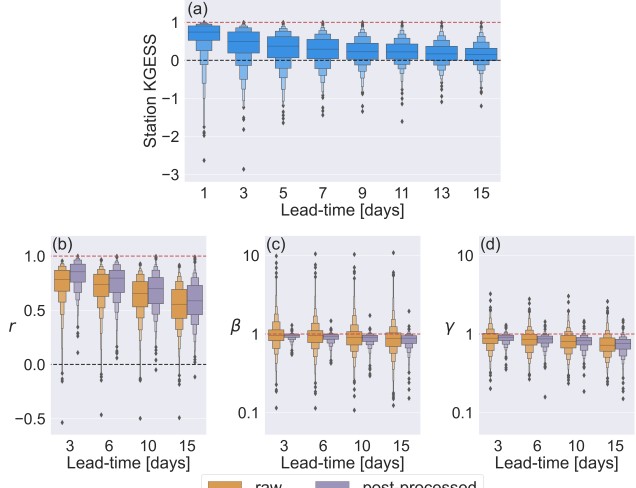

**Figure 5.** Comparison of the raw and post-processed forecast medians. (a) The Kling-Gupta Efficiency Skill Score (KGESS) for the forecast medians at all 522 stations for every other lead-time. Red dashed line shows the perfect score of KGESS $= 1$. Black dashed lines show KGESS value of 0. KGESS $> 0$ indicates the skill of the forecast median is improved by post-processing. KGESS $< 0$ indicates the skill of the forecast median is degraded by post-processing. The three components of the $KGE'$: (b) Correlation component, $r$. Black dashed line shows r $= 0$. (c) Bias ratio component, $\beta$. (d) Variability ratio component, $\gamma$. Red dashed lines show the perfect scores of 1 for all components. Both (c) and (d) have logarithmic y-axes.

degrades the forecast median increases with lead-time from 8% at 1 day to 26% at 15 days. However, the lowest KGESS values

become less extreme (i.e. not as negative). This increase in the KGESS of the most degraded stations is due to a decrease at longer lead-times in the skill of the raw forecast (used as the benchmark for the skill score) rather than an increase in the skill of the post-processed forecasts. This shows that the effect of naïve skill on the results should be considered; however, as the aim is to evaluate the impact of post-processing, it is appropriate to use the raw forecasts as the benchmark (Pappenberger et al., 2015b).

Figure 5b shows the correlation component, $r$, of the $KGE'$ for both the raw and the post-processed forecast medians. Post-processing improves the correlation between the forecast median and the observations for most stations, particularly at short lead-times where approximately 30% of stations have an improved correlation of over 0.1. However, at longer lead-times almost 40% of stations show a decrease in correlation with the observations and 10% show a decrease of more than 0.1 for at least one lead-time. The impact of post-processing on the correlation component of the $KGE'$ varies greatly between

stations. Notably, the flashiness of the catchment and whether or not the river is regulated can affect the performance of the post-processing (see Sect. 5.2.2). Additionally, the quality and length of the calibration time-series also have an effect (see Sect. 5.2.3).

Figure 5c shows the bias ratio, $\beta$, which indicates if on average the forecasts over or under-estimate the discharge at a station. The post-processing method attempts to correct for any consistent hydrological biases (differences between the observations





and the water balance simulation) during the hydrological uncertainty estimation part of the online correction (see Sect. 3.4.1). The climatological discharge distribution is conditioned on the discharge values from the recent period. The mean of the conditioned distribution is calculated in Eq. (14) as the mean flow of the observed time-series from the historic period (term 1) plus an amount dependent on the discharge values in the recent period (term 2). Therefore, assuming the mean flow does not change between the calibration (historic) and evaluation periods, any biases in the hydrological model climatology should be

corrected.

In general, the post-processing method does succeed in bias correcting the forecasts. For raw forecasts, the $\beta$-values range from approximately 10 (an over-estimation by an order of magnitude) to 0.1 (an under-estimation of an order of magnitude) with an almost equal split between over and under-estimation at short lead-times, and a tendency towards under-estimation at longer lead-times. For the post-processed forecasts the $\beta$-values are more tightly clustered around the perfect value of $\beta = 1$.

The largest improvements to the $\beta$-values are for stations where the flow is under-estimated by the raw forecasts. This is because many of the stations with raw $\beta$-values of greater than 1 are over-corrected such that the post-processed forecasts have $\beta$-values of less than 1. This is supported by the similarity of the median $\beta$-values for the raw and post-processed forecasts despite the decrease in the range of values. The over-correction is generally due to the under-estimation of high flows (see discussion on the third component of the $KGE'$, the variability ratio) which results in an under-estimation of the average flow

and hence a $\beta$-value of less than 1. For stations where the raw forecast largely over-estimates the discharge, this over-correction is not sufficient to cause a degradation to the forecast. However, for stations where the over-estimation is relatively small the over-correction can result in the post-processed forecasts being more biased than the raw forecasts.

In general, there is also a small decrease in the $\beta$-values at longer lead-times. This is present in both the raw and post-processed forecasts and is primarily caused by an increase in the under-estimation of high flows at longer lead-times as the skill

of the forecast decreases. However, for some stations the drift in $\beta$-values at longer lead-times is also caused by nonstationarity of the discharge distribution. A change in the mean flow from that of the calibration period (term 1 of Eq. (14)) means the hydrological uncertainty is calculated using an inaccurate climatological forecast. If the impact of the discharge values from the recent period (on which the climatological forecast is conditioned) is not large enough to counteract these inaccuracies, then this will result in a biased post-processed forecast. The magnitude of the impact of the recent discharge values decreases with

lead-time as the autocorrelation weakens. Therefore, any errors in the climatological forecast are more pronounced resulting in a more biased forecast at longer lead-times and a drift in the $\beta$-values.

Figure 5d shows the variability ratio, $\gamma$, which indicates if the forecast median is able to capture the variability of the flow. In general, the variability of the flow tends to be under-estimated by the raw forecast ($\gamma$ less than 1) because the magnitudes of the peaks relative to the mean flow are not predicted accurately particularly at longer lead-times. This results in a decrease of

the raw $\gamma$-values at longer lead-times. This drift is also visible in the post-processed $\gamma$-values. However, at all lead-times most stations show an improvement after post-processing (i.e. have a value of $\gamma$ closer to 1). Stations where the raw forecast over-estimates the variability ($\gamma$ above 1) are more likely to have the variability corrected by post-processing particularly at longer lead-times. At a lead-time of 15 days over 70% of these over-estimated stations have $\gamma$ values closer to 1 after post-processing.





The two main factors impacting the ability of the post-processed forecasts to capture the variability of the flow are 1) the

level of indication of the upcoming flow by the discharge values in the recent period, and 2) the spread of the raw forecast. In the Kalman filter the hydrological uncertainty distribution and the meteorological uncertainty distribution are combined to produce the final post-processed forecast (see Sect. 3.4.3). The impact that either distribution has on the final post-processed forecast is dependent on their relative spreads. The spread of the hydrological uncertainty is impacted by the discharge values in the recent period. Due to the skewedness of discharge distributions, the climatological forecasts tend to have a low probability

of high flows. If the recent discharge values show no indication of an upcoming high flow (e.g. no increase in discharge), the low probability of high flows is reenforced when the climatological forecast is conditioned on discharge values from the recent period (see Sect. 3.4.1). This decreases the spread of the hydrological uncertainty distribution and increases its weight within in the Kalman filter.

The meteorological uncertainty distribution is the spread corrected raw forecast and includes the variability due to the

meteorological forcings. For floods with meteorological drivers, if the magnitude of the peaks is under-predicted by the raw forecasts then the post-processed forecasts are also likely to under-predict the magnitude of the peaks. Alternatively, if the raw forecast is unconfident in the prediction of a peak (e.g. only a couple of members predict a peak) then it may not have a sufficient impact within the Kalman filter and the post-processed forecast may not predict the peak regardless of the accuracy of the ensemble members that do predict the peak. The impact of the spread correction is discussed further in Sect. 5.2.1.

**5.1.2 Timing of the peak discharge**

To evaluate the impact of post-processing on the ability of the forecast to predict the timing of the peak flow accurately the Peak-Time Error ($PTE$, see Sect. 4.3.2) is used. The aim of this assessment is to see how well the forecast is able to identify the time within the forecast period with highest flow and therefore greatest hazard. A $PTE$ of less than 0 indicates the peak is predicted too early whereas a $PTE$ of greater than 0 indicates the peak was predicted too late. Figure 6 shows the distribution

of the $PTE$ values for both the post-processed and raw forecasts for all forecasts where the maximum observed value in the forecast period exceeds the 90th percentile within the forecast period. The forecasts are split into three categories dependent on the lead-time at which the observed maximum occurs. Therefore, the distributions shown in each panel are truncated at different values of $PTE$. For example, an observation occurring on a lead-time between 1 and 5 days can at most be predicted 5 days early.

Approximately 50% of the forecast medians of the raw forecasts have no error in the timing of the peak for peaks that occur within lead-times of 1 to 5 days. This drops to 40% for post-processed forecasts. Both sets of forecasts have approximately 70% of forecasts with timing errors of 1 day or less. However, the post-processed forecasts tend to predict the peak too early whereas the raw forecasts are approximately equally split between predicting peaks too early and too late. Although more likely to correctly predict the timing of the peak, raw forecasts are also more likely than the post-processed forecasts to have larger

$PTE$ values indicating the peak is predicted several days too late. For maximum observed values occurring on lead-times of 6 to 10 days, the post-processed forecasts still tend to predict peaks earlier than the raw forecasts but are also more likely to predict the peak several days too late. For maximum observed values occurring at the longest lead-times of 10 to 15 days the





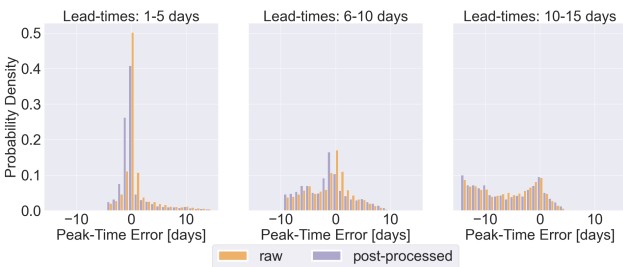

**Figure 6.** Histograms showing the probability distribution of Peak-Time Errors for all forecasts where the maximum observation is above the 90% percentile for the station (26807 forecasts) for raw forecasts (orange) and post-processed forecasts (purple). (a) Maximum observations occurs on lead-times of 1 to 5 days. (b) Maximum observations occurs on lead-times of 6 to 10 days. (c) Maximum observations occurs on lead-times of 10 to 15 days.

post-processed forecasts are slightly more likely to predict the timing of the peak correctly. However, they are also more likely to predict the peak too late. The distribution of $PTE$ values is bimodal for both sets of forecasts with a peak at $PTE = 0$ and

a peak at $PTE = -14$. This is usually due to the underestimation of the discharge at longer lead-times by the forecast median for forecasts that are preceded by high flows in the recent period resulting in high initial conditions.

Overall the impact of post-processing is small but tends towards the early prediction of the peak flow. However, there are three main limitations with this analysis. The first is that both sets of forecasts are probabilistic and therefore the median may not provide an adequate summary of the forecast. To account for the probability distribution, this analysis was repeated for the

lowest and highest percentiles (not shown). For the raw forecasts the lowest and highest ensemble members at each lead-time were used and for the post-processed forecasts the 1st and 99th percentiles were used (solid grey lines in Fig. 4). For both the lowest and highest percentiles the post-processed forecasts are more likely to predict the peak earlier (and usually too early) than the raw forecast regardless of lead-time.

Secondly, the evaluation here is forecast-based rather than peak-based in that the focus is the timing of the highest discharge

value in the forecast within the forecast period and not the lead-time at which a specific peak is predicted accurately. This was intentional as the twice-weekly production of the reforecasts means that a specific peak does not occur at each lead-time.

Finally, the combination of forecasts at all stations means the relationship between the runoff generating mechanisms and the $PTE$ cannot be assessed. It was found that although in general the post-processed forecasts predict the timing of the peak less accurately (too early), this degradation to the timing of the peak was more common if the maximum observation was

surrounded by relatively low observations than if the discharge was high for a longer period (not shown). The impact of slow and fast catchment responses is discussed in more detail in Sect. 5.2.2.

### 5.1.3   Threshold exceedance

The Relative Operating Characteristic (ROC) diagrams for the mean flow (MQ) and the mean annual maximum flow (MHQ) thresholds (see Sect. 4.3.3) are shown in Fig. 7. The diagrams show the probability of detection against the false alarm rate for





varying decision thresholds. The forecast period is split into three lead-time groups: 1-5 days, 6-10 days, and 11-15 days (see Sect. 4.3.3). The ROC scores for the MQ and MHQ thresholds are given in Table 1 for each lead-time group for the raw and post-processed forecasts, along with the corresponding skill scores (ROCSS). Both the raw and post-processed forecasts have ROC scores greater than 0.5 showing that they are more skilful than a climatological forecast.

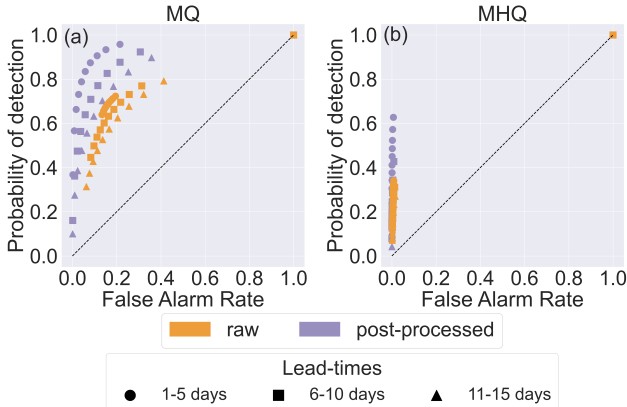

**Figure 7.** Relative Operating Characteristic diagrams for (a) the MQ threshold (118,888 observations above MQ), and (b) the MHQ threshold (2783 observations above MHQ). All stations are combined and groupings of lead-times are used (see Sect. 4.3.3.)

The spread of the raw forecasts is small at short lead-times. This is shown by the overlapping of the points in Fig. 7a for lead-times of 1-5 days (orange circles). The similarity of the points indicates that the decision thresholds are usually triggered simultaneously and therefore that the forecast distribution is narrow. The spread of the forecast increases with lead-time as the ensemble of meteorological forcings increases the uncertainty in the forecasts. Although the skill of the forecast median decreases with lead-time (see Sect. 5.1.1), the introduction of the meteorological uncertainty means the usefulness of the raw forecasts is similar for lead-times of 1-5 days and 6-10 days. This is shown by similarity of the ROC scores for these lead-time groups for the raw forecast.

Post-processing also accounts for the hydrological uncertainty allowing for a more complete representation of the total predictive uncertainty. In addition, as shown in Fig. 5c, post-processing bias corrects the forecast relatively well at short lead-times. The combination of spread and bias correction leads to an increase in the probability of detection for all but the highest decision thresholds and a decrease in the false alarm rate for almost all decision thresholds and lead-times. The added reliability gained from post-processing decreases with lead-time. The ROCSS for lead-times of 1-5 days at the MQ level is 0.8 but is only 0.45 for lead-times of 11-15 days.

The ROC diagram for the MHQ threshold (Fig. 7b) shows that the raw forecasts tend to cautiously predict high flows with the forecast much more likely to miss a flood than to issue a false alarm even for the lowest decision threshold. There is less improvement from post-processing than for the MQ threshold with the ROCSS for the MHQ threshold only reaching 0.48 for 1-5 days lead-time. For the MHQ threshold, the post-processing increases the probability of detection and decreases the false alarm rate at short lead-times. At longer lead-times the false alarm rate is still decreased by post-processing, but the probability





| Lead-time | MQ | | | MHQ | | |
|---|---|---|---|---|---|---|
| | ROC$raw$ | ROC$_{pp}$ | ROCSS | ROC$raw$ | ROC$_{pp}$ | ROCSS |
| 1-5 days | 0.78 | 0.96 | 0.87 | 0.68 | 0.83 | 0.48 |
| 6-10 days | 0.78 | 0.91 | 0.56 | 0.68 | 0.74 | 0.20 |
| 11-15 days | 0.76 | 0.87 | 0.45 | 0.67 | 0.69 | 0.08 |

**Table 1.** Relative Operating Characteristic Scores (ROCS) and corresponding skill scores (ROCSS) for the raw and post-processed (pp) forecasts for lead-times of 1-5 days, 6-10 days, and 10-15 days for the mean flow threshold (MQ) and the mean annual maximum threshold (MHQ).

of detection is also decreased for the largest decision thresholds. This reluctance to forecast larger probabilities also occurs with the MQ threshold and is due to the interaction between the hydrological and meteorological uncertainty in the Kalman filter discussed in Sect. 5.1.1.

745   Figure 8 shows reliability diagrams for the MQ and MHQ thresholds. For the MQ threshold (Fig. 8a) the raw forecasts are over-confident leading to under-estimation of low probabilities and over-estimation of high probabilities. The post-processed forecasts are more reliable but also tend to under-estimate low probabilities. The raw forecasts increase in reliability with lead-time whereas the reliability of the post-processed forecasts decreases. This is also true for the MHQ threshold.

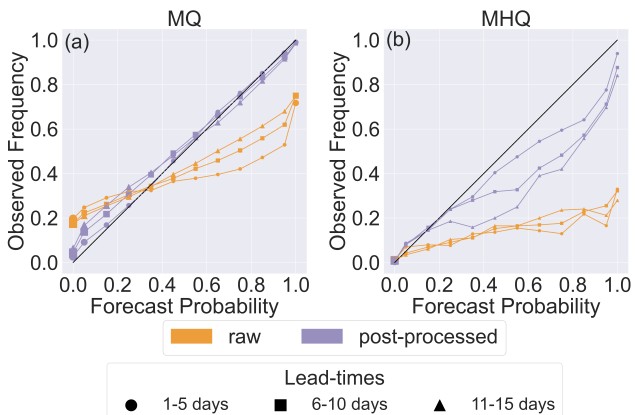

**Figure 8.** Reliability diagrams for (a) the mean flow threshold (MQ), and (b) the mean annual maximum flow (MHQ). All stations are combined and groupings of lead-times are used (see Sect. 4.3.3.)

Both sets of forecasts are consistently below the diagonal in the MHQ reliability diagram (Fig. 8b) indicating unconditional
750   biases. However, the post-processed forecasts have smaller biases consistent with the results discussed in Sect. 5.1.1. In addition, the raw forecast shows relatively poor resolution with events occurring at approximately the same frequency regardless of the forecast probability.





The distribution of forecasts (shown by marker size) is more uniform for the post-processed forecasts particularly at shorter lead-times. Since the ensemble reforecasts evaluated have 11 members and the operational forecasts have 73 members, the distribution for operational raw forecasts is expected to be slightly more even as the additional members allow for greater gradation in the probability distribution. The distribution of forecasts is skewed towards low probabilities showing similarly to the ROC diagrams (Fig. 7) that both sets of forecasts tend to cautiously forecast flows exceeding the MHQ threshold.

### 5.1.4 Overall skill

The Continuous Ranked Probability Skill Score (CRPSS) is used to evaluate the impact of post-processing on the overall skill of the probability distribution of the forecasts. Figure 9 shows the CRPSS for each station at lead-times of 3, 6, 10, and 15 days. Stations that are degraded by post-processing (CRPSS < 0) are circled in red. Stations that show a large increase in skill after post-processing (CRPSS > 0.9) are circled in cyan.

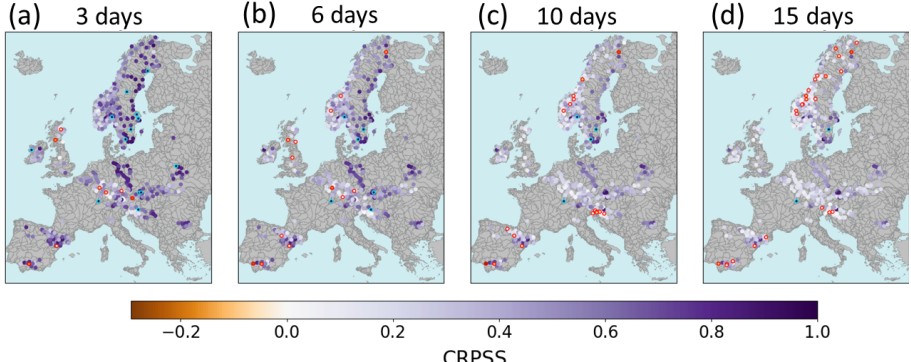

**Figure 9.** The Continuous Ranked Probability Skill Score (CRPSS) for all 522 stations for lead-times of 3, 6, 10, and 15 days. CRPSS values below 0 indicate the forecast probability distribution is on average less skillful after post-processing and values above 0 indicate added skill after post-processing. Markers are outlined in red if the CRPSS is below 0 and in cyan if the CRPSS is above 0.9.

As was seen with the KGESS for the forecast median, there is a decrease in the improvement offered by post-processing at longer lead-times. This can be seen in Fig. 9, by the gradual change from dark purple to light purple/white values for panels (a) to (d). It is also shown in the increase of red circles and the decrease of cyan circles. Approximately 55% of stations have a CRPSS of above 0.5 at a lead-time of 3 days and this decreases to 10% by a lead-time of 15 days. At a lead-time of 3 days, 8 stations are degraded by the post-processing and 13 stations have a CRPSS greater than 0.9. By a lead-time of 15 days these change to 24 degraded stations and only 2 stations with CRPSS values greater than 0.9. Many of the stations that are improved significantly have large hydrological biases. For example, one of the most improved stations at a lead-time of 15 days is in Rheinweiler, Germany (see Fig. 3) which has a large bias in the hydrological model output due to limitations in the representation of the drainage network in the model domain. The post-processing method can account for these biases (see Sect. 5.1.1) resulting in CRPSS values greater than 0.9 at all lead-times.





The lack of clustering of the stations with CRPSS values above 0.9 suggest that the magnitudes of the largest corrections are due to station dependent characteristic. On the other hand, the degraded stations at a lead-time of 3 days appear to cluster

in three loose regions. In all three regions the degradation is due to high short-duration peaks being captured better by the raw forecasts than the post-processed forecasts. At longer lead-times the Spanish catchments are still degraded but the Scottish stations are not. As discussed in Sect. 5.1.1 for the lowest KGESS values, this is due to a decrease in the skill of the raw forecasts. The degraded stations at lead-times of 10 and 15 days cluster in Spain, around the Kjolen Mountains, and in the Sava catchment. The poorly post-processed forecasts in the Sava catchment are downstream of a reservoir the impact of which is

discussed in Sect. 5.2.2.

Comparing the CRPSS values in Fig. 9 with the raw CRPS values shown in Fig. 3 shows similarities in the spatial pattern of the raw forecast skill and the spatial pattern of the magnitude of improvement due to post-processing. In general, stations with low CRPS scores (high skill) for the raw forecasts are improved the most by post-processing. For example, the west coast of the Scandinavian Peninsula has a lower raw skill in general and the level of improvement is also lower than that of the east

coast. However, there are some anomalies to this pattern. For example, the station in Cong Weir, Ireland has a relatively low raw forecast skill compared to surrounding catchments due to regulation of the streamflow but has a high CRPSS value at all lead-times (over 0.9 at 3 days). Additionally, whilst stations on the River Rhine and the River Oder have similar raw CRPS values the River Oder is improved more by post-processing. This suggests that post-processing is more effective at dealing with certain types of error and therefore that the benefit of post-processing is catchment dependent. This is discussed in Sect.

790 5.2.

As mentioned, many of the stations with CRPSS values below 0 at short lead-times are degraded due to peak flows being better predicted by the raw forecasts. Therefore, the skill of the forecast at different flow levels is evaluated. Figure 10 shows the distribution of CRPSS values for all stations evaluated over the 4 quartiles of discharge (Q1 lower quartile to Q4 upper quartile) such that each boxenplot contains 522 CRPSS values, one for each station evaluated over approximately 52 forecasts.

Only lead-times of 3, 6, 10, and 15 days are shown but these lead-times are representative of the results at similar lead-times.

The improvement for all 4 quartiles decrease with lead-time as has been seen previously in Fig. 5 and Fig. 9. The improvement from post-processing is smaller for higher flows. However, the majority of stations are still improved for these high flows with over 60% of stations being improved for discharge values in Q4 at a lead-time of 15 days. The high flows are often under-predicted by both sets of forecasts. As discussed in Sect. 5.1.1, the ability of the post-processed forecasts to capture the

magnitude of peaks is often determined by the relative spread of the hydrological and meteorological uncertainty distributions. Although Q4 is the category with the greatest number of degraded stations (CRPSS < 0), some stations are degraded more (have a lower CRPSS value) for discharge values in Q1. This is mainly due to the larger proportional errors for lower flows.

## 5.2 What impacts the performance of the post-processing method?

In the previous section the impact of post-processing was shown to vary greatly between stations. The following sections

investigate the factors that influence the effect of the post-processing method. The CRPSS is used in this analysis as it provides an assessment of the improvement or degradation to the overall skill of the probabilistic forecast.





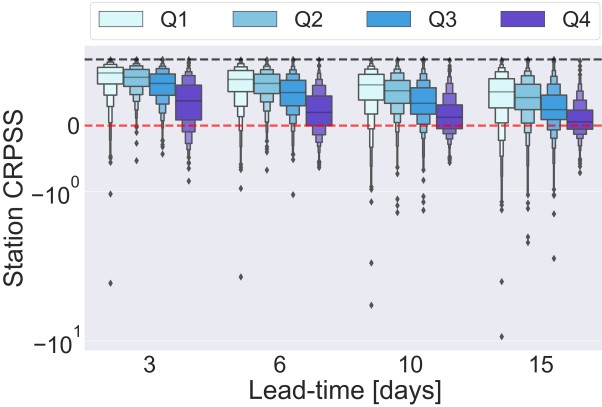

**Figure 10.** The Continuous Ranked Probability Skill Score (CRPSS) for all 522 stations calculated over the forecasts (approximately 52 forecasts) with flow values in the lowest quartile (Q1) to the highest quartile (Q4). CRPSS values below 0 indicate the forecast probability distribution is on average less skillful after post-processing and values above 0 indicate added skill after post-processing. A log-scale is used on the y-axis.

To aid the discussion of the key results some stations are highlighted. See Fig. 3 for the locations of the stations. Figure 11 shows the observed time-series (solid black line) for half the evaluation period (1 October 2017 to 30 September 2018) for six example stations; (a) Daldowie, Scotland. (b) Nytorp, Sweden. (c) Svarttjørnbekken, Norway. (d) Daugavpils, Latvia. (e) Porttipahta, Finland. (f) Montañana, Spain. The forecast median of the raw forecasts (oranges) and the post-processed forecasts (purples) are also plotted for lead-times of 3 days (circles), 6 days (crosses), and 15 days (triangles). These stations are discussed throughout Sect. 5.2 and were chosen as they allow some of the impacts of the post-processing to be visualised. Table 2 summaries the key results that each of the example stations highlight and all results are summarised in Sect. 6.

### 5.2.1  Type of uncertainty

This section looks at how meteorological and hydrological uncertainties affect the performance of the post-processing method. As mentioned in Sect. 1, the term meteorological uncertainties is used to refer to the uncertainty in the streamflow forecasts due to the error and uncertainty in the meteorological forcings, and not the error in the meteorological forecasts themselves. The magnitude of meteorological uncertainty is represented here by the CRPS of the raw ensemble forecast at each lead-time respectively. To remove the uncertainty due to the hydrological model, the water balance simulation is used as the "truth" value in the calculation of the CRPS, replacing the value of the observation, $y$, in Eq. (33). As both the forecast and the water balance simulation are produced using the same hydrological model, and the water balance simulation provides the initial conditions for the reforecasts, the only remaining uncertainty is from the forcings. The errors of the meteorological observations used to create the water balance simulation are considered negligible compared to those of the meteorological forecasts. The magnitude of the hydrological uncertainty is represented by the CRPS of the water balance simulation, with the observations used as the "truth" values, at each lead-time respectively. As both these values are deterministic the CRPS is equivalent to the square of the



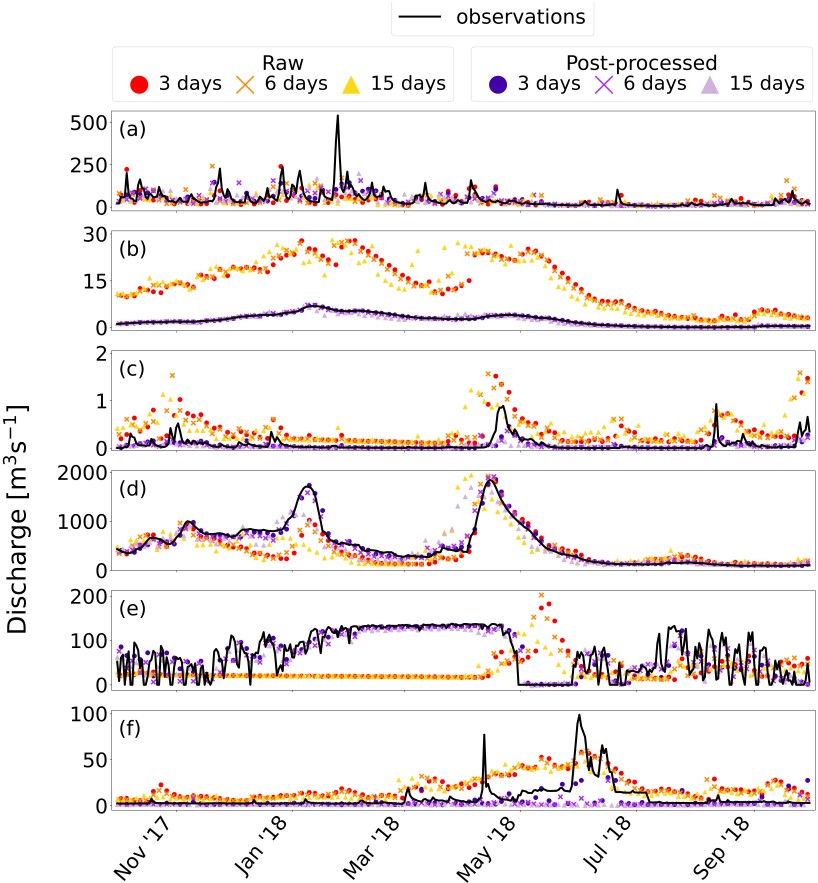

**Figure 11.** Observation time-series for one year of the evaluation period from October 2017 to October 2018 for 6 example stations. The forecast medians of the raw and post-processed forecasts are shown for lead-times of 3, 6, and 15 days. (a) Daldowie, Scotland. (b) Nytorp, Sweden. (c) Svarttjørnbekken, Norway. (d) Daugavpils, Latvia. (e) Porttipahta, Finland. (f) Montañana, Spain.

absolute difference between the two values. Both metrics, for the meteorological and hydrological uncertainties, are averaged over all 208 forecasts for each station. So that the errors are comparable between catchments they are calculated in terms of specific discharge ($mm\ d^{-1}$) instead of discharge ($m^3 s^{-1}$).

Figure 12 shows density plots of the CRPSS values for all stations vs the hydrological errors (a-c) and meteorological errors (d-f) for lead-times of 6, 10, and 15 days. A lead-time of 3 days is not shown here as the meteorological forcings have often not had a significant effect on the forecasts resulting in small distribution of meteorological errors across stations. However, the relationships discussed below are present at all lead-times. The 15 stations with the largest hydrological errors at each lead-time have been removed from the main analysis because these stations appear to show a slightly different pattern as shown in Fig. 12g and discussed below.





| Panel | Station | Description of key results | Section |
|---|---|---|---|
| (a) | Daldowie, Scotland | – Meteorological errors are not corrected as well as hydrological errors.<br>– Poor post-processing of peaks for flashy catchments. | 5.2.1, 5.2.2 |
| (b) | Nytorp, Sweden | – Large biases due to limitations of the drainage network are well corrected. | 5.2.1, 5.2.2 |
| (c) | Svarttjørnbekken, Norway | – Post-processing is beneficial for stations where the hydrological model is uncalibrated. | 5.2.2 |
| (d) | Daugavpils, Latvia | – Slowly responding catchments benefit from post-processing the most.<br>– Post-processing can account for poor modelling of slow hydrological processes such as snowmelt. | 5.2.2 |
| (e) | Porttipahta, Finland | – Regulated catchments benefit from post-processing. | 5.2.2 |
| (f) | Montañana, Spain | – The quality of the calibration time-series is more important than the length of the time-series. | 5.2.3 |

**Table 2.** Key results and the section that provide more information for each of the six stations used as examples and for which time-series are shown in Fig. 11

The purple lines in Fig. 12 show the least-squares regression line of best fit for the relationship between the CRPSS vs the hydrological and meteorological errors. In general, an increase in either the hydrological or meteorological uncertainties, decreases the improvement due to post-processing. However, this relationship is much stronger for the meteorological errors, which account for over 13% of the variability observed in the CRPSS values whereas the hydrological errors only account for around 1% of this variability. This suggests that hydrological errors are better corrected by the post-processing method and

hence their magnitude has little impact on the performance of the post-processing method. The EMOS method is used to correct the spread of the raw forecast to account for the meteorological uncertainty (see Sect. 3.4.2). However, the EMOS method is not used to correct for any bias introduced by the meteorological forcings as is sometimes done (e.g. Gneiting et al., 2005; Hemri et al., 2015b; Zhong et al., 2020). Both bias and spread correction are performed for the hydrological uncertainties. In Sect. 5.1.4 it was noted that the forecasts for the Rhine and the Oder catchments have similar skill but the Oder was improved





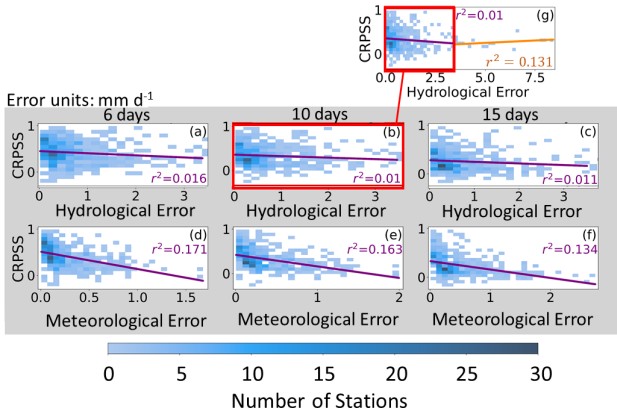

**Figure 12.** Density plots showing the station CRPSS for lead-times of 6 days (a and d), 10 days (b and e), and 15 days (c and f), against hydrological error (a-c) and meteorological error (d-f). The largest 15 hydrological errors are excluded from panels a-f. See Sect. 5.2.1 for an explanation of the metrics used to represent the hydrological and meteorological errors. Purple lines show the line-of-best-fit calculated using linear regression and the associated $r^2$ are given within each panel. (g) The CRPSS against hydrological error including the 15 largest hydrological errors for a lead-time of 10 days. The orange line shows the line-of-best-fit for the station with large hydrological errors.

more. It was found (not shown) that this is because the errors in the raw forecast of the Rhine were mainly meteorological but those of the Oder were mainly hydrological.

Although the $r^2$ values are small some trends are observed in their variation with lead-time. The relationship between the meteorological errors and the CRPSS value is slightly stronger at shorter lead-times. This is partly because the EMOS spread correction parameters are lead-time invariant. The spread of the raw forecast tends to be very small at short lead-times, because

all ensemble members have the same initial conditions, but increases as the differing meteorological forcings propagate through the catchment system. Multiplying the spread at all lead-times by a constant value means the spread retains this structure. Therefore, at shorter lead-times the meteorological forcings are more influential within the Kalman filter than at longer lead-times (as discussed in Sect. 5.1.1). On the one hand, if the raw forecast is skilled at short lead-times then this greater influence is beneficial and may, for example, allow the post-processed forecast to predict an upcoming peak. On the other hand, any

large errors contained in the raw forecasts propagate through to the post-processed forecasts. Meteorological errors are due to the inaccuracy or the lack of confidence of the raw forecast and can impact the post-processed forecast as discussed in Sect. 5.1.1. For example, the largest peak in the time-series for the station in Daldowie, Scotland (see Fig. 11a) is not predicted by the raw forecast medians and it is therefore not predicted by the post-processed forecast medians either because no information about the precipitation-driven upcoming peak has been provided by the raw forecast. As discussed in Sect. 5.1.1, the spread of

the meteorological uncertainty distribution (see Sect. 3.4.2) relative to the spread of the hydrological uncertainty distribution (see Sect. 3.4.1) determines the weight of each within the Kalman filter and hence their impact on the post-processed forecast. Figure 11a shows that many of the peaks at the Daldowie station in winter 2017/2018 are forecast accurately by the raw forecast median but are not forecast by the post-processed forecast. This suggests that the hydrological uncertainty distribution is most





impactful in the Kalman filter. The observations in the recent period often don't indicate an upcoming flood so the conditioning
on these observations results in a hydrological uncertainty distribution which confidently, but incorrectly, predicts a low flow.
The hydrological uncertainty distribution has a larger weighting in the Kalman filter due to its confidence which results in
the information of the upcoming flow provided by the meteorological uncertainty distribution being ignored. This ignoring of
the meteorological information is also the reason for the poorly post-processed forecasts for some stations in Spain (see Fig.
9). As some Mediterranean catchments have very low hydrological variability except for rare large peaks, it is unlikely that
observations from the recent period will indicate an upcoming peak, and therefore, the hydrological uncertainty distribution
usually predicts a low flow with very high confidence. As this will often lead to the meteorological forcings being ignored,
even though extreme precipitation can be an important runoff generating mechanism (Berghuijs et al., 2019), post-processed
forecasts for these catchments should be used cautiously particularly when the raw forecasts predict a flood.

For the hydrological errors the $r^2$ values for lead-times of 10 and 15 days are similar whereas the $r^2$ value for a lead-time of
6 days is slightly higher. In general, the $r^2$ value decreases for lead-times of 1 day to approximately 6 days (not shown) and for
lead-times longer than 6 days the $r^2$ values remain at approximately 0.01. This suggests that forecast dependent errors due to
the initial conditions and the interaction of the meteorological forcings in the hydrological model are corrected at shorter lead-
times, but at longer lead-times the correction is mainly to consistent hydrological model errors. The consistent hydrological
errors can be corrected well regardless of magnitude.

The 15 stations with the largest hydrological uncertainties is show a small increase in average CRPSS with increasing
hydrological uncertainties. This trend is visualised by the orange line in Fig. 12g but the limited number of data points makes
the calculation irresolute. The relationship is only shown here for a lead-time of 10 days but is present at all longer lead-times.
Most of the hydrological uncertainty in these cases is caused by large consistent biases rather than forecast dependent errors.
These consistent biases are usually easier to correct resulting in higher CRPSS values when the bias of the raw forecasts is
larger. For example, one of the 15 stations with the largest hydrological uncertainties is the station in Nytorp, Sweden which has
a large bias in the raw forecasts (see Fig. 11b). As discussed in 5.1.1 the post-processing method is able to correct for consistent
biases by conditioning the climatological forecast on the discharge values in the recent period. This results in post-processed
forecasts that much more closely follow the observations as shown in Fig. 11b, and large KGESS and CRPSS values at all
lead-times.

## 5.2.2 Catchment characteristics

The catchments within the EFAS domain vary greatly in terms of size, location, and flow regime. This section discusses
catchment characteristics that impact the performance of the post-processing method namely: upstream area, response time, el-
evation, and regulation. In Fig. 13, box-and-whisker plots are used to show the distribution of the CRPSS values for all stations
at every other timestep with the whiskers extending to the 5th and 95th percentiles. The stations are split into categories de-
pending on (a) the size of the upstream area, (b) the time of concentration, and (c) the elevation. Values for these characteristics
are extracted from static LISFLOOD maps used operationally.



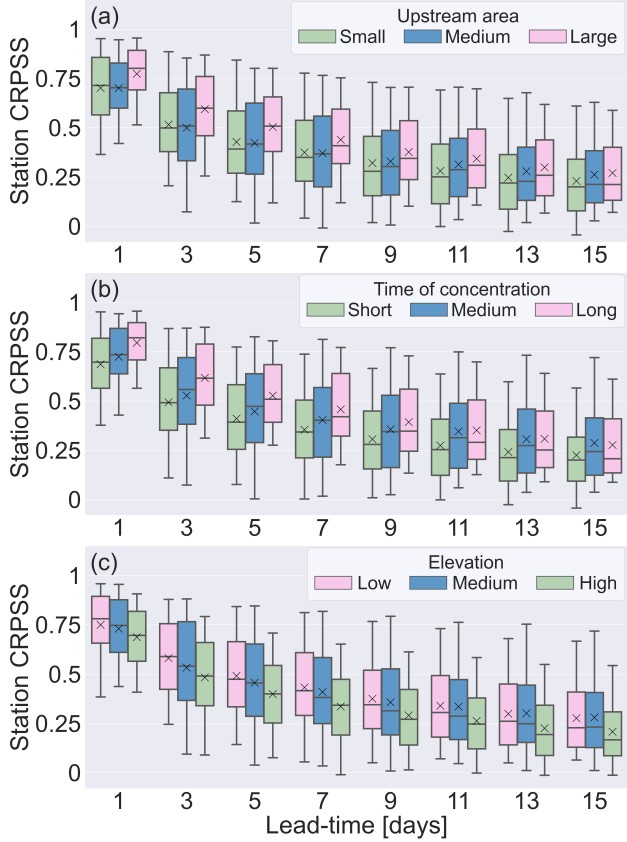

**Figure 13.** The CRPSS for all 522 stations at every other lead-time with stations categorised by their catchment characteristics. (a) Upstream area. Small catchments: less than 1000 $km^2$ (165 stations), Medium catchments: between 1000 $km^2$ and 5000 $km^2$ (204 stations), Large catchments: larger than 5000 $km^2$ (153 stations). (b) Time of concentration. Fast response catchments: less than 24 hours (253 stations), Medium response catchments: between 24 and 48 hours (144 stations), Slow response catchments: more than 48 hours (126 stations). (c) Elevation Low-elevation catchments: less than 150 $m$ (178 stations), Medium-elevation catchments: between 150 $m$ and 400 $m$ (168 stations), High-elevation catchments: more than 400 $m$ (177 stations)

Figure 13a shows that in general large catchments (larger than 5000 $km^2$) are improved more by post-processing than medium (between 1000 $km^2$ and 5000 $km^2$) and small (less than 1000 $km^2$) catchments. The difference is largest at shorter lead-times and decreases until near negligible at a lead-time of 15 days. The relationship between medium and small catchments is less consistent. At short lead-times the median CRPSS value for small catchments is higher than for medium catchments but for longer lead-times the converse is true. However, it was found that by removing stations with an upstream area smaller than 500 $km^2$ (henceforth referred to as very small catchments) from the analysis the remaining small stations (with upstream areas between 500 $km^2$ and 1000 $km^2$) are in general improved less by post-processing than medium catchments at all lead-times. This results in a single trend: that in general post-processing improves forecasts more for larger catchments. The need





to remove very small catchments from the analysis to clearly identify this trend is due to 1) uncalibrated stations and 2) errors in the upstream areas. Both reasons complicate the relationship between upstream area and CRPSS values for very small catchments and are discussed below.

Most stations with upstream areas (provided by local authorities) smaller that 500 $km^2$ were not included in the calibration of LISFLOOD for EFAS 4 (Mazzetti et al., 2021). Whilst for some very small catchments the uncalibrated model may have
good hydrological skill, for others the exclusion from calibration may result in larger hydrological errors than for calibrated catchments. As discussed in Sect. 5.2.1 hydrological errors are well-corrected by post-processing, therefore resulting in larger CRPSS values for some very small uncalibrated catchments than for larger calibrated catchments where the errors may be primarily meteorological. For example, the station in Svarttjørnbekken, Norway has a catchment area provided by local authorities of 3.4 $km^2$ and was therefore not included in the calibration. Fig. 11c shows that the raw forecasts over-estimate the
variability of the flow resulting in large over-estimations of the magnitude of the peaks. This is likely due to the lack of model calibration at this station.

The minimum area increment of the LISFLOOD static map used to categories the stations is the area of one grid-box, 25 $km^2$. Therefore, the upstream areas are multiples of 25 $km^2$ and thus may not match those provided by local authorities. The station in Svarttjørnbekken Norway is also impacted by this over-estimation of the upstream area. In LISFLOOD its upstream
area of 3.4 $km^2$ is rounded to 25 $km^2$ (over 7 times the size of the catchment). As shown in Fig. 11c, this over-estimation of the upstream area results in a consistent bias in the raw forecast even at low flows. The post-processed forecasts are bias corrected as discussed in Sect. 5.1.1. Errors in the upstream area occur for stations of all sizes but in general are proportionally larger for very small catchments.

In Fig. 13b the time of concentration is used to represent the catchment response time. Stations are split into fast response
catchments (response times of less than less than 24 hours), moderate response catchments (between 24 and 48 hours), and slow response catchments (more than 48 hours). At short lead-times, slowly responding catchments outperform medium and fast responding catchments. Since large catchments tend to have slower responses, this suggests response time is partly responsible for the greater improvement experienced by large catchments. Slower responses result in stronger autocorrelations therefore the relationship between discharge values in the recent period and those in the forecast period are stronger. Therefore allowing
greater improvements for slowly responding catchments than for flashier catchments. This is shown by comparing the time-series of the Daugavpils station (Fig. 11d) which has a time of concentration of approximately 195 hours with that of the Daldowie station (Fig. 11a) which has a time of concentration of 27 hours. The Daugavpils station has a slow response with peaks lasting two months (longer than the length of the recent period) whereas the Daldowie station responds faster with peaks only lasting a week at most (shorter than the length of the forecast period). As such the post-processing method can correct
forecasts much better for the Daugavpils station. Additionally, as mentioned in Sect. 2, the forecasts have daily timesteps and the forecast variable is the average discharge over the preceding 24 hours. For fast response catchments this means that the entire duration of the catchment response may be contained within a single timestep which can makes peak-time errors more damaging to the forecast. In Sect. 5.1.2 the post-processed forecasts were found to have larger timing errors than the





raw forecasts. This is likely to have contributed to the worse performance of the post-processing method for fast response
catchments compared to slow response catchments.

Figure 13b shows at longer lead-times the improvements to medium and slowly responding catchments are similar but the
fast response catchments remain least improved. However, most stations still benefit from being post-processed even at lead-
times larger than their time of concentration. This is useful as operationally there is a delay in the availability of observations
(both discharge observations and meteorological observations used to create the water balance simulation) whereas here it is
assumed that all observations up to the production time of the reforecast are available. Therefore, these results suggest that
although the CRPSS may be smaller there is still an operational benefit to post-processing.

In Fig. 13c catchments are categorised by the height of the station above sea level: low-elevation catchments (less than 150
$m$), medium-elevation catchments (between 150 $m$ and 400 $m$), and high-elevation catchments (more than 400 $m$). At all
lead-times catchments at higher elevations are improved less than lower-lying catchments. This is partly due to mountainous
catchments tending to have faster response times. Additionally, as shown in Sect. 5.2.1, the post-processing is sensitive to
meteorological errors. Precipitation forecasts in mountainous regions can be biased due to insufficient resolutions to represent
the orography in the NWP systems (Lavers et al., 2021; Haiden et al., 2014). Alfieri et al. (2014) found that when compared
to the water balance simulation (i.e. equivalent to the metric for meteorological error used here) the raw ensemble forecasts
are negatively biased in mountainous regions due to an under-estimation of the precipitation. In addition to under-estimation
of the precipitation amount, spatial and temporal errors in the meteorological forecast can lead to errors in the streamflow
forecasts. Spatio-temporal errors are more damaging for small catchments (Pappenberger et al., 2011) and tend to increase
with lead-time (Haiden et al., 2014, 2021b). Since catchment area tends to decrease at higher elevations this could account for
the slight increase in the difference in CRPSS values between high-elevation catchments and lower-lying catchments at longer
lead-times. The effect of station elevation on the performance of the post-processing method explains the cluster of degraded
stations around the Kjolen Mountains (see Fig. 9).

The regulation of rivers via reservoirs and lakes is difficult to model. Raw forecasts for many regulated catchments were
found to have negative correlation with the observations (not shown) including the Porttipahta station in Finland which had the
worst raw correlation of all stations. The improvement due to post-processing at regulated stations is dependent on whether
the reservoir is in the same state during the recent and forecast periods and hence whether the discharge values from the recent
period provide useful information about the state of the reservoir. In this study, a station is considered regulated if it is within 3
grid-boxes downstream of a reservoir or lake in the LISFLOOD domain or if data providers have reported that the station is on
a regulated stretch of the river. Figure 14 shows the CRPSS values of the 42 regulated stations (black lines) and the distribution
of the CRPSS values of the unregulated stations (green distribution) for lead-times of 3, 6, 10, and 15 days. The distribution
for the unregulated stations is estimated using kernel density estimation with the dashed line showing the median value and the
dotted lines showing the interquartile range. The mean CRPSS values are indicated by crosses of the respective colours.

At all lead-times, the CRPSS values of most regulated stations are above the median of the unregulated stations. Additionally,
the mean CRPSS value of the regulated stations is at least 0.1 higher than that of the unregulated stations for all lead-times
longer than 1 day. At short lead-times the difference is smaller because the recent discharge values are similarly informative





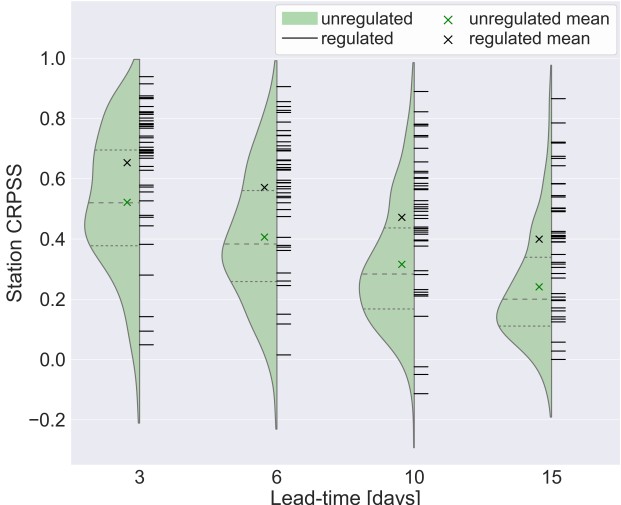

**Figure 14.** Violin plot of the CPRSS values for the 480 unregulated stations (green distribution) and the 42 regulated catchments (black lines) at lead-times of 3, 6, 10, and 15 days.

for both sets of stations. The largest difference in mean CRPSS values is at a lead-time of 6 days. By this lead-time the recent

discharge values are less relevant for unregulated catchments. However, if the reservoir is in the same state as during the recent period, then the recent observations remain relevant for regulated catchments allowing for greater improvement. At longer lead-times it becomes more likely that the reservoir will have changed state and therefore that the information provided by the recent discharge values is not useful. For example, the Porttipahta station in Finland is located at the Porttipahta reservoir and its time-series is shown in Fig. 11e. In May 2018 the discharge is $0 \ m^3 s^{-1}$ for approximately a month. The raw forecast

does not capture this decrease in discharge, but the post-processed forecast median is very accurate even at longer lead-times. However, at the start and end of this zero-flow period the post-processed forecasts do not perform as well for a lead-time of 15 days (purple triangles). This is because the reservoir has changed state since the forecast production time and therefore the recent observations are not providing correct information about the state of the reservoir. The Porttipahta has a correlation component of the $KGE'$ of approximately -0.5 at all lead times for the raw forecast but post-processing increases this to

between 0.91 and 0.68 at all lead-times.

If the reservoir changes state often the skill of the post-processed forecasts is less consistent. Small but regular controlling of the flow via reservoirs can create poorly performing post-processed forecasts. For example, the post-processed forecasts are not able to capture the relatively rapid oscillation in the autumn and winter months of the Porttipahta time-series as well as they do the month long zero-flow period in the summer (see Fig. 11e). It is thought that small but regular regulation is partly

responsible for the cluster of degraded stations on the River Sava shown for a lead-time of 10 days in Fig. 9c. Three of the degraded stations in this cluster are regulated and are the three regulated stations with the lowest CRPSS values at all lead-times shown in Fig. 14.





It is interesting to consider whether other hydrological processes that are difficult to model can be accounted for by post-processing. For example, the peak in the winter and spring in the Daugavpils catchment (see Fig. 11d) is largely dominated by snow and ice melt (Škute et al., 2008). Snowmelt is known to be a difficult process to model (Alfieri et al., 2014). As shown in Fig. 11d the raw forecasts do not predict the large peak in late January but the post-processed forecasts are able to capture the peak due to their conditioning on recent observations which include the increase in discharge due to snowmelt. Similar results were seen in other catchments with snow dominated regimes. Although the identification of dominating runoff generating mechanisms for all catchments and seasons is beyond the scope of this study, the results presented in this section suggest that post-processing can correct for errors introduced by the imperfect modelling of slow hydrological processes.

### 5.2.3 Calibration time-series

The length of the historic period, and therefore the length of the time-series used to calibrate the station model, varies between stations. The maximum length is dictated by the corresponding water balance simulation which is available from 1 January 1990. However, many stations have shorter time-series due to the availability of observations. The lengths of the calibration time-series of the evaluated stations vary from 2 to 27 years with a mean of 21 years and a median of 23 years. This distribution is heavily skewed towards longer time-series compared to the distribution for all operationally post-processed stations. Operationally only 45% of stations have time-series greater than 20 years and 25% have time-series of less than 5 years whereas within the evaluated stations 70% have time-series over 20 years and only 5% have time-series less than 5 years. The difference in the distribution of time-series lengths is due to the criteria used to select the stations for evaluation. Many of the stations with short time-series have an overlap between the evaluation and calibration periods and therefore cannot be used.

Figure 15 shows the CRPSS values for each lead-time with stations split by the length of their calibration time-series into unequally sized categories (see caption): very short time-series (up to 15 years), short time-series (between 15 and 20 years), medium time-series (between 20 and 25 years), and long time-series (over 25 years). These categories were chosen to investigate the impact of the length of the calibration time-series whilst keeping the number of stations in each category as large as possible. These initial comments ignore the very short time-series (green) which are discussed in more detail below. For lead-times up to 3 days, long time-series in general lead to more improvement by post-processing than shorter time-series. For lead-times from 3 to 7 days there is little difference between the performance of the post-processing at stations with medium times-series (pink) and long time-series (yellow) but the performance is lower for stations with short time-series. Longer time-series allow the joint distribution between the observations and the water balance simulation to be more rigorously defined allowing a more accurate conditioning of the forecast on the discharge values from the recent period. For lead-times greater that 7 days the CRPSS distributions for all categories are similar. As discussed in Sect. 5.2.1, post-processing corrects forecast specific errors at short lead-times but at longer lead-times it is mainly consistent errors to the climatology that are corrected. The similarity of the CRPSS distributions suggests that short time-series are sufficient to capture these consistent errors. As the calibration time-series of the evaluated stations tend to be longer than that of the average operationally post-processed station, these results suggest that the average improvement at short lead-times may be over-estimated. However, as the evaluated





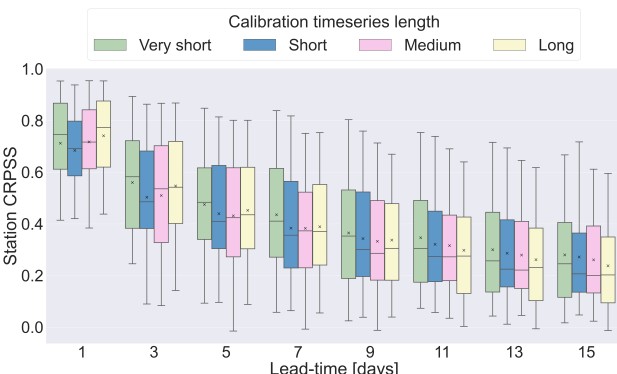

**Figure 15.** The CRPSS for all 522 stations at every other lead-time with stations categorised by the length of their calibration time-series. Very short time-series: less than 15 years (63 stations), Short time-series: 15 to 20 years (93 stations), Medium time-series: 20 to 25 years (119 stations), Long time-series: over 25 years (247).

stations are also operationally post-processed these results are still indicative of the improvement expected operationally at some stations.

At the longest lead-times there is a small decrease in the CRPSS for longer time-series. This slight trend may be due to the different sizes of the categories, but it may also suggest that longer time-series are more likely to include non-stationary

patterns. There are several reasons that this can occur at a station. A non-exhaustive list includes changes to the rating curve due to river channel alterations, changes to the joint distribution due to catchment modifications, and errors in the observations due to instrument failure. The station in Montañana (shown in Fig. 11f) is an example of a station where a period of poor quality observations in the calibration time-series impact the calibration resulting in a large jump in the CDF of the observed discharge distribution as highlighted by a red circle in Fig. 16b. This CDF is used in the NQT and the large jump results in non-smooth

forecast probability distributions. Additionally, these errors were found to impact the estimation of the joint distribution. The Montañana station has the lowest correlation value after post-processing (see Fig. 5) due to a degradation in the correlation of approximately 0.7. This may be due to the poorly defined joint distribution. This example highlights the importance of the quality of the observation data especially in short time-series where each observation has more impact on the estimated distributions. Even stations with time-series of less than 5 years are corrected for both consistent biases and forecast dependent

errors by post-processing. This suggests that the priority should be to use the best quality data available even if that means that the calibration time-series is short.

The category for stations with very short time-series (green) does not follow the patterns discussed above and instead performs relatively well in terms of median CRPSS at all lead-times. In general shorter time-series tend to be more recent and so benefit not only from improved river gauging technology but also because non-stationarity between the calibration and

evaluation period is less likely to be an issue. Although a full sensitivity analysis is beyond the scope of this study, these results suggest that very short time-series can be used if necessary to define an accurate joint distribution although longer, high-quality time-series are preferable.


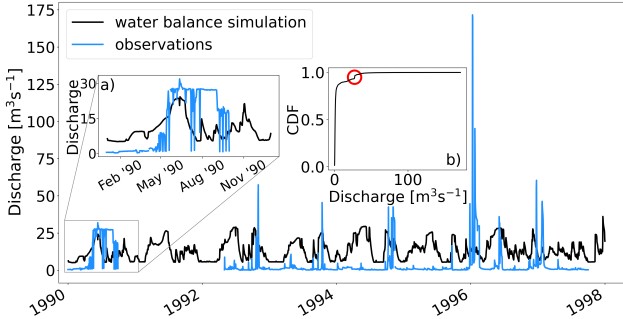

**Figure 16.** Observations (blue) and water-balance simulation (black) time-series used in the calibration of the station model for the station in Montañana. a) Section of the calibration time-series with errors in the observations. b) the Cumulative Distribution Function (CDF) of the observed discharge distribution calculated during the calibration. Red circle indicates a jump in the CDF due to the section of the time-series shown in a).

## 6 Conclusions

Post-processing is a computationally efficient method of quantifying uncertainty and correcting errors in streamflow forecasts.
Uncertainties enter the system from multiple sources including the meteorological forcings from numerical weather prediction systems (here referred to as meteorological uncertainties), and the initial hydrological conditions and hydrological model (here referred to as hydrological uncertainties). The post-processing method used operationally in the European Flood Awareness System (EFAS) uses a method motivated by the Ensemble Model Output Statistics (Gneiting et al., 2005) method to account for the meteorological uncertainty and the Multi-Temporal Model Conditional Processor (Coccia, 2011) to account for the hydrological uncertainty. The EFAS domain includes catchments of varying characteristics for which the same post-processing method is used. In this paper we used reforecasts to investigate the added skill gained by post-processing and how these improvements vary across the domain. This study aimed to answer two research questions.

First, does the post-processing method provide improved forecasts? Our results show that for the majority of stations the post-processing improves the skill of the forecast with median Continuous Ranked Probability Skill Scores (CRPSS) of between 0.74 and 0.2 at all lead-times. This improvement is greatest at shorter lead-times of up to 5 days but post-processing is still beneficial up to the maximum lead-time of 15 days. The bias and spread correction provided by the post-processing increases the reliability of the forecasts and increased the number of correctly forecast flood events without increasing the number of false alarms. However, the post-processed forecasts also led to the flood peak often being forecast too early by approximately a day. Although, forecasts for floods events at most stations did benefit from post-processing the greatest improvements were to forecasts for normal flow conditions.

Second, what affects the performance of the post-processing method? Several factors were found to impact the performance of the post-processing method at a station. The post-processing method is more easily able to correct hydrological errors than meteorological errors. This is mainly because no bias-correction is performed for the meteorological errors whereas



hydrological errors are bias corrected by conditioning the forecast on the recent observations. Therefore, stations where the errors were primarily due to hydrological errors were improved more. As the hydrological errors tend to be larger than the meteorological errors this is beneficial; however, more research is required to fully account for biases due to the meteorological forcings as well.

The post-processing method was found to easily account for consistent hydrological biases that were often due to limitations in the model representation of the drainage network. However, the correction of forecast specific errors (due to initial conditions and meteorological forcings) was largely determined by the response time of the catchment. Therefore, the greatest improvement was seen in catchments larger than $5000\ km^2$ and catchments less than $100\ m$ above sea level as these catchments tended to have longer response times. Additionally, post-processing was able to correct for errors due to difficult to model hydrological processes, such as regulation and snowmelt, when recent observations contained relevant information about the discharge.

The use of long historic observational time-series for the offline calibration is beneficial particularly for correcting forecast specific errors. However, time-series shorter than 15 years were found to be sufficient for correcting consistent errors in the model climatology even at a lead-time of 15 days. The quality of the observations in the historic time-series is important and errors in the time-series degraded the performance of the post-processing method and limit the usefulness of the forecasts.

These results highlight the importance of post-processing within the forecasting chain of large-scale flood forecasting systems. They also provide a benchmark for end-users of the EFAS forecasts and show the situations when the post-processed forecasts can provide more accurate information that the raw forecasts. These results also highlight possible areas of improvement within the EFAS and the factors that must be considered when designing and implementing a post-processing method for large-scale forecasting systems.

*Code and data availability.* The raw reforecasts (https://doi.org/10.24381/cds.c83f560f) and the water balance simulation (https://doi.org/10.24381/cds.e3458969) are available from the Copernicus Climate Data Store. The post-processed forecasts and evaluation code are available from the University of Reading Research Data Archive (http://dx.doi.org/10.17864/1947.333).

*Author contributions.* GM: Conceptualization, data curation, formal analysis, investigation, methodology, project administration, software, validation, visualization, writing - original draft preparation, writing - review editing, CB: data curation, resources, software, HC: Conceptualization, Funding acquisition, methodology, project administration, supervision, writing – review editing, SD: Conceptualization, Funding acquisition, methodology, project administration, supervision, writing – review editing, TJ: Resources, software, CM: Resources, writing – review editing, CP: Conceptualization, Funding acquisition, methodology, project administration, supervision, writing – review editing.

*Competing interests.* The authors declare that they have no conflict of interest.





*Acknowledgements.* We gratefully acknowledge the financial support provided by the following: Gwyneth Matthews is funded in part by the EPSRC DTP 2018-19 University of Reading (EP/R513301/1) and ECMWF. Sarah Dance was funded in part by the UK EPSRC DARE project (EP/P002331/1). Hannah Cloke acknowledges funding from UK Natural Environmental Research Council (NERC): The Evolution of
Global Flood Risk (EVOFLOOD) Project Grant NE/S015590/1. We thank Paul Smith for the development and operational implementation of the post-processing method. We are grateful for the advice provided by Shaun Harrigan and David Richardson. We thank members of the Water@Reading research group for their advice and support.



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
