# Peer review of "Evaluating the impact of post-processing medium-range ensemble streamflow forecasts from the European Flood Awareness System"

_Hydrology and Earth System Sciences, 2021_

## Author Comment (AC1)

**Response to RC1 for hess-2021-539: Matthews, G., et al. Evaluating the impact of post-processing medium-range ensemble streamflow forecasts from the European Flood Awareness System**

We thank the reviewer for their thoughtful comments and suggestions which we believe will greatly improve the manuscript. The reviewer's comments have been summarised and numbered for clarity. The authors responses are in blue. Comments regarding typos or grammatical mistakes are discussed last.

General comments:
1. "The discussion of the results is, however, a bit too long and could be summarized more concisely."
We thank the reviewer for this suggestion. If invited to revise the paper, we will shorten this section. Please see the responses to comments 5, 6, and 24 which explain how this will be done.

2. [Section 2] "There are many EFAS papers available now and all describe the EFAS system. This could be shortened and only the differences to the operational settings of EFAS should be explained, in particular how the reforecasts are used."
We thank the reviewer for this suggestion. We will shorten the material currently in section 2 by focusing on the details that are different between the operational forecasts and the reforecasts and removing the additional details. For example, we will remove the details of the NWP systems and direct the reader to the relevant sources but will discuss that the operational forecasts are multi-model ensembles whereas the reforecasts are from a single model. To avoid repetition, we will move some material from the current section 4.2.1 that discusses the reforecasts to section 2.

3. "This small number of members causes difficulties in computing the CRPS for making a fair comparison with a CRPS derived from the PDFs of the post-processed forecasts (see for example Zamo and Naveau, 2018). However, this problem is not mentioned and the presented results of the CRPSS should be treated with care."
We thank the reviewer for this comment and for highlighting the relevant literature. If invited to revise the paper, we will add the following to section 4.3.4: "It should be noted that the error in the calculation of the CRPS for the raw ensemble forecasts is likely to be large compared to that of the post-processed forecasts because of the limited number of ensemble members (Zamo and Naveau, 2018). However, as this evaluation is of the post-processing method no corrections to account for the ensemble size are made (e.g. Ferro et al., 2008) as the impact of the post-processing would be difficult to differentiate from that of the CRPS correction."

4. "A most recent paper from Skøien, et al. (2021) has a similar topic about evaluating the post-processing methods for EFAS (EMOS and the application of transformations like NQT). Therefore, the differences and novelty of this study should be stressed clearly and discussed in more detail."
We thank the reviewer for bringing this relevant article to our attention. The Skøien et al. (2021) study differs from our study in several ways: (a) Skøien et al. research the implementation of a new post-processing method whereas the post-processing method that we use is the operational post-processing method used in the EFAS system. Therefore, our results are of interest to end-users of the EFAS forecast products as well as the hydrological research community. (b) The focus of Skøien et al. is to compare different methods with a focus on forecast features whereas we compare a single method across many catchments with a focus on catchment characteristics. (c) The combination of the MCP and the EMOS methods allows us to account for the errors in the hydrological model as well as in the meteorological forcings whereas Skøien et al. did not investigate the hydrological model uncertainty. (d) Skøien et al. use lead-time dependent EMOS parameters whereas we use lead-time invariant parameters. We will discuss points a and b in the section 1 and points c and d in section 5.2.1.

5. "Although the analysis of the different aspects like catchment size, elevation, regulation, length of period, are very interesting, it maybe could be shortened focusing on floods, which is the main topic of EFAS."
We thank the reviewer for this suggestion. As more end users are using EFAS forecasts, particularly the post-processed forecast products for which no flood alerts are issued, for information about non-extreme streamflow situations as well as for floods, we believe that the evaluation at all flow levels is beneficial. However, we will

shorten sections 5.2.1-5.2.3 by restructuring each section to remove repetition. For example, lines 1028-1032 and lines 1043-1045 make similar remarks so we will combine these comments.

6. "Also, the detailed analysis of the Nash-Sutcliffe (KGE) is maybe too long, since the results of the post-processing methods are probabilistic and the KGE reduces the information content to the mean (median) of the Ensembles (or PDF)."
We will shorten section 5.1.1 by removing some of the specific quantitative detail. For example, on line 622 we will remove "where approximately 30\% of stations have an improved correlation of over 0.1". We will also remove repetitive descriptions of Fig. 5 (e.g. line 620). Additionally, we will reduce the discussion on the drift in β-values with lead-time by focusing only on the impact of non-stationarity.

Specific comments:
7. [Page 5, section 2] "Highlighting the differences between the operational setting of EFAS and the setting used in this study should be sufficient. More details can be found in many other papers."
Please see the response to comment 2.

8. "However, the calibration period of the LISFLOOD model is missing. Is there an overlap between the period for calibrating the parameters of the hydrological model and the historical period p for the off-line calibration?"
The calibration of LISFLOOD was performed over the period 1990 – 2017 using 6-hourly data where available (Mazzetti et al., 2021). Therefore, there is an overlap with the historic period, p. We will add this information to Section 2.

9. [Figure 1] "the index of the parameters μ and Σ is ψ, whereas in the caption you use the index Φ."
We thank the reviewer for highlighting this inconsistency. If invited to revise the paper, we will change the index of parameters μ and Σ in the caption of Figure 1 to ψ.

10. "You mention several times (e.g. line 172, page 7) that the minimum for the off-line calibration is 2 years. However, for the fitting of the GPD you use 1000 values (page 9). So you will need more than 2 years?"
If the length of the calibration timeseries for a station is only two years (less than 1000 data points) then the station is still calibrated but all data points will be tested as the break point. We will add this clarification to Section 3.3.1.

11. "I would suggest including a list of nomenclature, so you avoid repetitive descriptions like the tilde for the physical space (line 182, 204, 220, 340) and the definition of the timestep notation introduced on page 7 (lines 176-177)."
We will remove the repetitive descriptions and direct the readers to the notation section (Section 3.1).

12. [Line 172, page 7] "you write that the location parameter a is used for defining the breakpoint, but in Figure 2 the shape parameter c is used as breakpoint."
We thank the reviewer for highlighting this inconsistency. We will change the variable name for the breakpoint in Figure 2 to $a$.

13. [Line 251, page 9] "you write about consistence between the 2 distributions. What does it mean? How do you check this?"
To create a smooth discharge distribution there must be no step change in the distribution at the breakpoint between the kernel density estimation and the Generalised Pareto Distribution (GPD). We will replace line 251 with "The scale parameter, b, is determined analytically by the constraint that the density distribution must be equal at the breakpoint for both the GPD and the KDE distributions."

14. [Line 257, page 10] "the concentrated likelihood method is mentioned without any further explanation what this method does. Maybe some more details would be helpful."
The concentrated likelihood method requires one parameter, say $a$, to be expressed as a function of the other parameters, say $b$ and $c$. The likelihood of $a$ can then be calculated given the values of $b$ and $c$. We will explain the concentrated likelihood method and how it is applied within the method.

15. [Lines 261-262] "Also, it is not clear for me how the GPD is weighted"

Given that the total probability density must equal 1, the weighting of the kernel density part of the distribution is the value of the cumulative distribution function at the breakpoint, denoted F(a) and the weighting of the GPD part of the distribution is then 1-F(a). We will change lines 261-262 to "The likelihood function for the full distribution is the product of the likelihood function of the kernel density and the likelihood function of the GPD weighted by their contribution to the total density, F(a) and 1-F(a), respectively (MacDonald et al., 2011)."

16. [Lines 270-280, page 11] "Whereas the description of the linear approximation is maybe not necessary."
We thank the reviewer for this suggestion. We will remove the detailed description of the linear approximation.

17. [Page 13] "it is not clear for me why you have observations (line 336) and water balance (line 337) for the period until t, but the forecasts (line 339) until t-1?"
We thank the reviewer for highlighting this ambiguity. The forecast that is produced at time *t* is the forecast that we are post-processing. Therefore, although observations and water balance values are available (in this idealised situation) for time *t* and can be used in the post-processing method, the forecast at time *t* cannot. To clarify this issue we will change line 335 to "As well as the current forecast produced at time t, $x_t(t+1 : t+T)$, the online correction requires the following input data from the recent period:".

18. [Line 411, page 16] "you write that a set of forecasts are used to estimate the two spread correction parameters. How did you choose the size of these sets of forecasts?"
The size of the set of forecasts is the number of forecasts available within the recent period. The length of the recent period, q=40, was determined using tuning experiments that were performed prior to the work of this study (Paul Smith, personal communication, September 25, 2020). We will explain this in the introduction to Section 3.4.

19. [Figure 3] "In Figure 3 you write CRPS besides the legend bar, but it should be CRPSS?"
Figure 3 shows the CRPS of raw reforecasts with the water balance used as the "true" value. We use this metric as a measure of the meteorological error which is discussed within the results in Section 5.1.1. We will sign-post the reader to this results section to motivate the use of the CRPS.

20. [Line 483, page 19] "you write that only 11 reforecasts are available (I suppose this 11 comes from 40 days/7day x 2). The number 11 could be misleading, since it happens to coincide with the number of members mentioned in the next sentence"
We thank the reviewer for highlighting this ambiguity. We will replace line 483 with: "Whereas operationally daily forecasts for each day of the recent period are available, here only two reforecasts are available for each week of the recent period reducing the number of forecasts used to estimate the EMOS parameters from 40 to 11."

21. "Why do you fix it [q] to 40 days? Since there is this a discrepancy between the operational setting and this analysis anyhow, you could set q to a longer period to include more reforecasts (e.g. q=70 ~ 20 reforecasts)."
We thank the reviewer for this comment. We chose not to extend the recent period to account for the reduced number of reforecasts to avoid seasonal variations in the EMOS parameters. will explain this in section 4.2.1.

22. [Line 486] "I don't understand why the mean discharge value is predicted for the previous 6 hours?"
In the EFAS 4 system the hydrological model, LISFLOOD calculates the discharge as the average over the previous 6 hours for both the water balance simulations and the forecasts. However, the post-processing is currently only performed at a daily timestep. Therefore, the 6-hourly modelled discharge values must be aggregated to a daily timestep. We will add this explanation to Section 4.2.1.

23. [Figure 11a] "The difference between the raw and the post-processed forecasts in Fig. 11a (mentioned on page 35, line 862) is very difficult to see and almost not visible."
We thank the reviewer for highlighting this issue. Line 862 discusses two peaks which are forecast by the raw forecasts but not the post-processed forecasts. If invited to revise the paper, we will highlight these two peaks using boxes (see Figure 1 below).

[Figure]

*Figure 1: Modified version of Figure 11a. The grey boxes highlight the two peaks discussed on line 862.*

24. [Page 41] "the paragraph from line 1004 – 1010 can be removed"
If invited to revise the paper, we will remove this paragraph.

25. "I have some doubts about your suggestion that very short periods are sufficient (line 1045): the chance that such a short period will show the variability of the discharge needed for applying the NQT is rather small and the fitting of the GPD almost impossible. Consequently, the back-transformation of the variables from the Normal space will always produce poor and very unreliable results for floods."
We thank the reviewer for this comment. We will change the wording at line 1045 to "Although a full sensitivity analysis is beyond the scope of this study, these results suggest that very short time-series can be used, if necessary, to correct for consistent biases, although longer time-series are preferable. However, care should be taken when forecasting high flows since a short timeseries will not allow for a robust calculation of the upper tail of the discharge distribution (see Sect. 3.3.1) which will likely cause errors in the forecast probability distribution (Bogner et al., 2012)."

26. "The citation of Coccia (line 1135) is incomplete."
We thank the reviewer for this suggestion. We will correct this reference.

27. "Also, the term "Multi-Temporal" in combination with the MCP (MT-MCP) is mentioned only in line 153-154 and in the conclusions (line 1054), but is not explained."
We thank the reviewer for highlighting this missing explanation. We will add the following to the introduction (line 64): "The method discussed in this study is partially motivated by the Multi-Temporal Model Conditional Processor (MT-MCP, Coccia, 2011) which extends the original MCP method for application to multiple lead-times simultaneously."

28. Grammatical errors and typos
   a.  "Line 349: …using a MCP method …"
   b.  "Page 14, line 372: in the recent period …"
   c.  "Line 880: …uncertainties show a small increase"
   d.  "Line 1021  ..greater than.."
We thank the reviewer for highlighting these errors. We will correct these errors.

References:

Zamo, M., Naveau, P. Estimation of the Continuous Ranked Probability Score with Limited Information and Applications to Ensemble Weather Forecasts. Math Geosci **50,** 209–234 (2018). https://doi.org/10.1007/s11004-017-9709-7)

Ferro, C.A.T., Richardson, D.S. and Weigel, A.P. (2008), On the effect of ensemble size on the discrete and continuous ranked probability scores. Met. Apps, 15: 19-24. https://doi.org/10.1002/met.45

Skøien, J. O., Bogner, K., Salamon, P., & Wetterhall, F. (2021). On the Implementation of Postprocessing of Runoff Forecast Ensembles, Journal of Hydrometeorology, 22(10), 2731-2749

Mazzetti, C., Decremer, D., Prudhomme, C. Challenges of the European Flood Awareness System (EFAS) hydrological calibration. Poster presented at: Joint Virtual Workshop on "Connecting global to local hydrological modelling and forecasting: scientific advances and challenges"; June 29, 2021; Online [Available: https://events.ecmwf.int/event/222/overview].

MacDonald, A., Scarrott, C. J., Lee, D., Darlow, B., Reale, M., and Russell, G.: A flexible extreme value mixture model, Computational Statistics & Data Analysis, 55, 2137–2157, 2011.

Bogner, K., Pappenberger, F., and Cloke, H.: The normal quantile transformation and its application in a flood forecasting system, Hydrology and Earth System Sciences, 16, 1085–1094, 2012.

Coccia, G.: Analysis and developments of uncertainty processors for real time flood forecasting, Ph.D. thesis, alma, http://amsdottorato.unibo.it/3423/, 2011.

---

## Author Comment (AC2)

**Response to RC2 for hess-2021-539: Matthews, G., et al. Evaluating the impact of post-processing medium-range ensemble streamflow forecasts from the European Flood Awareness System**

We thank the reviewer for their helpful comments and believe that they will make the evaluation more rigorous and useful. The reviewer's comments have been numbered for clarity. The authors responses are in blue.

Full review:

1. "All aspects presented are of interest, however I do wonder whether the paper could be separated into two more focussed manuscripts, perhaps one focussing on the novel aspects of the post-processing method and validating its assumptions, and a second on evaluation the benefits and investigating factors that influence its performance."
We thank the reviewer for this suggestion. Although we had considered separating the paper into two papers, we prefer to keep the manuscript as one paper, as we feel it is important that the methods are discussed alongside their practical performance. However, we plan to shorten the paper (see response to RC1, particularly comments 5 and 6).

More specific comments:

2. "The sample covariance matrix is used to characterise the joint distribution of the historic observations and water balance simulations, equation 7. There are potential issues that may be encountered using this approach and it would be good understand whether special treatments have been needed to overcome these."

   i. "The covariance matrix is computed over a set of historic observations and is likely to have inflated, or spurious, correlations over long lags if the seasonal cycle of streamflow is not considered."
Spurious correlations are not treated in the current method and the joint distribution is naively assumed to be consistent throughout the year. The main reason for this is that many stations do not have a sufficiently long timeseries to consider seasonal distributions. If invited to revise the paper, we will add this information to Section 3.3.2.

   ii. "For large sample covariance matrices such as those estimated in this study, missing observations can lead covariance matrices that are not positive definite."
The covariance matrices are adjusted using a minimum eigenvalue threshold approach to guarantee they are positive definite. We will include this process in Section 3.3.2.

3. "The KGE analysis is performed using the median as a point estimate of the forecast ensemble. The results obtained for the post-processed forecasts, particularly the bias ratios and variability ratios of less than one at long lead times, are not unexpected as the variance of the forecast median will be considerable more damped that the mean. The forecast mean is likely to be a better choice as the point estimate of the forecast ensemble. Some theoretical justification of the use of the ensemble mean with measures of squared error can be found in Gneiting (2011)."
We thank the reviewer for this suggestion. We have now performed the calculations of the modified KGE for the ensemble mean. Figure 1a (below) shows the modified KGESS for the ensemble median (blue) and the ensemble mean (green) and shows that the distribution of the KGESS is similar for both point estimates. Additionally, Figures 1b, 1c, and 1d show the components of the modified KGE for the ensemble median (orange), post-processed forecast (purple), and the ensemble mean (cyan). As we can see in Fig 1b and Fig. 1d, the distributions of the correlations and variability ratios are similar for the ensemble mean and the ensemble median. In Fig. 1c we see that the median of the distribution

of bias ratios (shown by the central black line) for the ensemble mean (cyan) is higher at all lead-times than those of the ensemble median and the post-processed forecast.

In the original manuscript, the ensemble median was chosen because operationally the ensemble forecasts are often represented by boxplots where the median at each timestep is shown. We believe that the comparison of the ensemble median with the median of the post-processed forecast is a useful evaluation for end-users who may be choosing between the two products. Therefore, we will add text motivating the choice of the ensemble median in Section 4.3.1. However, we would also briefly discuss the ensemble mean and include the lower panels of Figure 1 (1b, 1c, and 1d) in the supplementary material.

[Figure]

Figure 1: Kling-Gupta Efficiency analysis for the raw ensemble median, post-processed median, and the raw ensemble mean. (a) KGE skill score with the post-processed forecast as the benchmark, (b) correlation coefficient, (c) bias ratio, (d) variability ratio.

4. "In this paper, the analysis of peak timing is conditioned on observations exceeding a threshold (90th percentile discharge threshold) within the forecast period and is likely to result in a biased evaluation of forecasts. A more rigorous approach would be to select the events based on forecasts exceeding the threshold."

We thank the reviewer for highlighting this limitation. We have run the analysis again using a forecast exceedance threshold and note that the bimodal distribution that was shown for lead-times 10-15 days is no longer present (see Figure 2 below). We believe the bimodal nature was due to forecasts that failed to predict an event being more harshly penalised than forecasts that predicted an event that did not occur. We will change the peak-time error analysis to use a forecast exceedance threshold.

5. "I also believe that rather than evaluating the timing of the peak in the forecast median, which doesn't correspond to the peak in any individual hydrograph, a more representative point estimate of the forecast timing error would be to compare the median (or mean) time to peak across all ensemble members to the timing of the observed peak. "

We thank the reviewer for this suggestion. We have performed the analysis with both the peak of the ensemble median and the median of the peaks of each of the ensemble members and found that the distributions are very similar. We have rerun the analysis using the criteria of forecast exceedance (see reply to comment 4). Figure 2 (below) shows the peak-time error for the peak of the ensemble median of the raw forecast (orange), the post-processed median (purple) and the median of the peaks for all ensemble members of the raw forecast (green). The two distributions calculated from the raw ensemble forecasts are similar in comparison with the post-processed forecast with the raw median performing slightly better at longer lead-times. Therefore, we will keep the comparison between the post-processed forecast and the ensemble median, but we will use the exceedance of the forecasts as the event criteria (see reply to comment 4).

6. "line 373 - values in the recent perion should be "values in recent period"
We thank the reviewer for highlighting this mistake. We will correct this mistake.

7. "Line 825 - CRPS calculated on deterministic forecasts is equivalent to the absolute error not the square absolute error."
We will also correct this mistake.

8. "Figures - The size of multi-panel figures (e.g. Figure 9, 12) could be increased to better illustrate the detail"
We thank the reviewer for highlighting the figures which are unclear. We will increase the size of Figures 5, 6, 9, and 12.

[Figure]

Figure 3: Peak-Time Error analysis using forecast exceedance as the event criteria for the peak of the raw ensemble median(orange), the peak of the post-processed forecast median (purple), and the median value of the peak timing of the ensemble members (green)

---

## Author Response (AR1)

**Response to reviewers' comments for hess-2021-539:** Matthews, G., et al. **Evaluating the impact of post-processing medium-range ensemble streamflow forecasts from the European Flood Awareness System**

We thank the editor and reviewers for their thoughtful comments and suggestions which we believe have greatly improved the manuscript and made the evaluation more useful. We respond to Reviewer Comment 1 and Reviewer Comment 2 in order. The reviewers' comments have been summarised and numbered for clarity. The authors responses are in blue. Comments regarding typos or grammatical mistakes are discussed last for each review. Line numbers refer to the lines in the revised manuscript.

**RC1**

General comments:
1. "The discussion of the results is, however, a bit too long and could be summarized more concisely."
We thank the reviewer for this suggestion. We have shortened this section. Please see the responses to comments 5, 6, and 24.

2. [Section 2] "There are many EFAS papers available now and all describe the EFAS system. This could be shortened and only the differences to the operational settings of EFAS should be explained, in particular how the reforecasts are used."
We thank the reviewer for this suggestion. We have shortened section 2 by focusing on the details that are different between the operational forecasts and the reforecasts and removing the additional details. For example, we have removed the details of the NWP systems and directed the reader to the relevant source. To avoid repetition, we have moved some material from section 4.2.1 that discusses the reforecasts to section 2.

3. "This small number of members causes difficulties in computing the CRPS for making a fair comparison with a CRPS derived from the PDFs of the post-processed forecasts (see for example Zamo and Naveau, 2018). However, this problem is not mentioned and the presented results of the CRPSS should be treated with care."
We thank the reviewer for this comment and for highlighting the relevant literature. We have added the following to section 4.3.4 (lines 572-576): "It should be noted that the error in the calculation of the CRPS for the raw ensemble forecasts is likely to be large compared to that of the post-processed forecasts because of the limited number of ensemble members (Zamo and Naveau, 2018). However, as this evaluation is of the post-processing method no corrections to account for the ensemble size are made (e.g. Ferro et al., 2008) since the impact of the post-processing would be difficult to differentiate from that of the CRPS correction."

4. "A most recent paper from Skøien, et al. (2021) has a similar topic about evaluating the post-processing methods for EFAS (EMOS and the application of transformations like NQT). Therefore, the differences and novelty of this study should be stressed clearly and discussed in more detail."
We thank the reviewer for bringing this relevant article to our attention. The Skøien et al. (2021) study differs from our study in several ways: (a) Skøien et al. research the implementation of a new post-processing method whereas the post-processing method that we use is the operational post-processing method used in the EFAS system. Therefore, our results are of interest to end-users of the EFAS forecast products as well as the hydrological research community. (b) The focus of Skøien et al. is to compare different methods with a focus on forecast features whereas we compare a single method across many catchments with a focus on catchment characteristics. (c) The combination of the MCP and the EMOS methods allows us to account for the errors in the hydrological model as well as in the meteorological forcings whereas Skøien et al. did not investigate the hydrological model uncertainty. (d) Skøien et al. use lead-time dependent EMOS parameters whereas we use lead-time invariant parameters. We have discussed points a and b in section 1 and points c and d in section 5.2.1.

5. "Although the analysis of the different aspects like catchment size, elevation, regulation, length of period, are very interesting, it maybe could be shortened focusing on floods, which is the main topic of EFAS."
We thank the reviewer for this suggestion. As more end users are using EFAS forecasts, particularly the post-processed forecast products for which no flood alerts are issued, for information about non-extreme streamflow situations as well as for floods, we believe that the evaluation at all flow levels is beneficial. However, we have shortened sections 5.2.1-5.2.3 by restructuring each section to remove repetition.

6. "Also, the detailed analysis of the Nash-Sutcliffe (KGE) is maybe too long, since the results of the post-processing methods are probabilistic and the KGE reduces the information content to the mean (median) of the Ensembles (or PDF)."

We have shortened section 5.1.1 by removing some of the specific quantitative detail. We have also removed repetitive descriptions of Fig. 5. Additionally, we have reduced the discussion on the drift in β-values with lead-time by focusing only on the impact of non-stationarity.

Specific comments:
7. [Page 5, section 2] "Highlighting the differences between the operational setting of EFAS and the setting used in this study should be sufficient. More details can be found in many other papers."
Please see the response to comment 2.

8. "However, the calibration period of the LISFLOOD model is missing. Is there an overlap between the period for calibrating the parameters of the hydrological model and the historical period p for the off-line calibration?"
The calibration of LISFLOOD was performed over the period 1990 – 2017 using 6-hourly data where available (Mazzetti et al., 2021). Therefore, there is an overlap with the historic period, p. We have added this information to Section 2 lines 126-127 and Section 4.2.2 lines 484-486.

9. [Figure 1] "the index of the parameters μ and Σ is ψ, whereas in the caption you use the index Φ."
We thank the reviewer for highlighting this inconsistency. We have changed the index of parameters μ and Σ to ψ in the caption of Figure 1.

10. "You mention several times (e.g. line 172, page 7) that the minimum for the off-line calibration is 2 years. However, for the fitting of the GPD you use 1000 values (page 9). So you will need more than 2 years?"
If the length of the calibration timeseries for a station is only two years (less than 1000 data points) then the station is still calibrated but all data points will be tested as the break point. We have added "If there are less than 1000 data points (i.e. $p<1000$) then all data points are tried as the location parameter." to section 3.3.1, lines 242-243.

11. "I would suggest including a list of nomenclature, so you avoid repetitive descriptions like the tilde for the physical space (line 182, 204, 220, 340) and the definition of the timestep notation introduced on page 7 (lines 176-177)."
We have removed the repetitive description regarding the tilde and direct the readers to the notation section (Section 3.1). We have removed some of the repetition of the time step notation. However, we have retained the reminder to the reader on line 279, as we feel this is helpful to aid the reader's understanding.

12. [Line 172, page 7] "you write that the location parameter a is used for defining the breakpoint, but in Figure 2 the shape parameter c is used as breakpoint."
We thank the reviewer for highlighting this inconsistency. We have changed the variable name for the breakpoint in Fig. 2 to *a*.

13. [Line 251, page 9] "you write about consistence between the 2 distributions. What does it mean? How do you check this?"
To create a smooth discharge distribution there must be no step change in the distribution at the breakpoint between the kernel density estimation and the Generalised Pareto Distribution (GPD). We have clarified this on lines 245-247: "The scale parameter, $b$, is determined analytically by the constraints that the density distribution must be equal at the breakpoint for both the GPD and the KDE distributions, and the integral of the full density distribution function with respect to discharge must be equal to 1"

14. [Line 257, page 10] "the concentrated likelihood method is mentioned without any further explanation what this method does. Maybe some more details would be helpful."
We have added "The parameters of the GPD are determined using the concentrated likelihood method (see steps 4-6 below). The concentrated likelihood method allows the maximum likelihood estimates of multiple parameters to be determined by first expressing one parameter in terms of the others (Takeshi, 1985)." to section 3.3.1.

15. [Lines 261-262] "Also, it is not clear for me how the GPD is weighted"
We have added a clarification on lines 253-254: "The full distribution is the combination of the KDE and GPD weighted by their contribution to the total density, $F_\eta(a)$ and $1-F_\eta(a)$, respectively (MacDonald et al., 2011)."

16. [Lines 270-280, page 11] "Whereas the description of the linear approximation is maybe not necessary."
We thank the reviewer for this suggestion. We will remove the detailed description of the linear approximation.

17. [Page 13] "it is not clear for me why you have observations (line 336) and water balance (line 337) for the period until t, but the forecasts (line 339) until t-1?"
We thank the reviewer for highlighting this ambiguity. The forecast that is produced at time $t$ is the forecast that we are post-processing. To clarify this issue, we have added lines 325-326: "As shown in Fig. 1, as well as the current forecast produced at time $t$, the online correction requires the following input data from the recent period:".

18. [Line 411, page 16] "you write that a set of forecasts are used to estimate the two spread correction parameters. How did you choose the size of these sets of forecasts?"
The size of the set of forecasts is the number of forecasts available within the recent period. The length of the recent period, q=40, was determined using tuning experiments that were performed prior to the work of this study (Paul Smith, personal communication, September 25, 2020). We have explained this in the introduction to Section 3.4.

19. [Figure 3] "In Figure 3 you write CRPS besides the legend bar, but it should be CRPSS?"
Figure 3 shows the CRPS of raw reforecasts. The relationship between the raw forecast skill and the improvement gained by post-processing is discussed in section 5.1.4. We have corrected this in the caption of Fig. 3.

20. [Line 483, page 19] "you write that only 11 reforecasts are available (I suppose this 11 comes from 40 days/7day x 2). The number 11 could be misleading, since it happens to coincide with the number of members mentioned in the next sentence"
We thank the reviewer for highlighting this ambiguity. We have clarified this on lines 468-470: "Whereas operationally daily forecasts for each day of the recent period are available, here only two reforecasts are available for each week of the recent period. This reduces the number of forecasts used to estimate the EMOS parameters from 40 to 11."

21. "Why do you fix it [q] to 40 days? Since there is this a discrepancy between the operational setting and this analysis anyhow, you could set q to a longer period to include more reforecasts (e.g. q=70 ~ 20 reforecasts)."
We thank the reviewer for this comment. We have added "We did not extend the recent period to maintain consistency with the operational system and to avoid introducing errors due to any seasonal variation in the EMOS parameters." to section 4.2.1 (lines 470-472).

22. [Line 486] "I don't understand why the mean discharge value is predicted for the previous 6 hours?"
In the EFAS 4 system the hydrological model, LISFLOOD calculates the discharge as the average over the previous 6 hours for both the water balance simulations and the forecasts. We have added this explanation to Section 2. However, the post-processing is currently only performed at a daily timestep. Therefore, the 6-hourly modelled discharge values must be aggregated to a daily timestep. We have added this explanation to Section 4.2.1.

23. [Figure 11a] "The difference between the raw and the post-processed forecasts in Fig. 11a (mentioned on page 35, line 862) is very difficult to see and almost not visible."
We thank the reviewer for highlighting this issue. Lines 831-833 discuss two peaks which are forecast by the raw forecasts but not the post-processed forecasts. We have now highlighted these two peaks using boxes.

24. [Page 41] "the paragraph from line 1004 – 1010 can be removed"
We have removed this paragraph.

25. "I have some doubts about your suggestion that very short periods are sufficient (line 1045): the chance that such a short period will show the variability of the discharge needed for applying the NQT is rather small and the fitting of the GPD almost impossible. Consequently, the back-transformation of the variables from the Normal space will always produce poor and very unreliable results for floods."

We thank the reviewer for this comment. We have replaced this on lines 849-853 with "Although a full sensitivity analysis is beyond the scope of this study, these results suggest that very short time-series can be used, if necessary, to correct for consistent biases, although longer time-series are preferable. However, care should be taken when forecasting high flows since a short timeseries will not allow for a robust calculation of the upper tail of the discharge distribution (see Sect. 3.3.1) which will likely cause errors in the forecast probability distribution (Bogner et al., 2012)."

26. "The citation of Coccia (line 1135) is incomplete."

We thank the reviewer for this comment. We have corrected this reference.

27. "Also, the term "Multi-Temporal" in combination with the MCP (MT-MCP) is mentioned only in line 153-154 and in the conclusions (line 1054), but is not explained."

We thank the reviewer for highlighting this missing explanation. We have added the following to the introduction (line 64-66): "The method discussed in this study is partially motivated by the Multi-Temporal Model Conditional Processor (MT-MCP, Coccia, 2011) which extends the original MCP method for application to multiple lead-times simultaneously."

28. Grammatical errors and typos
   a. "Line 349: …using a MCP method …"
   b. "Page 14, line 372: in the recent period …"
   c. "Line 880: …uncertainties show a small increase"
   d. "Line 1021  ..greater than.."

We thank the reviewer for highlighting these errors. We have corrected these errors (lines: 340, 361, 954, and 1108).

**RC2**

Full review:

1. "All aspects presented are of interest, however I do wonder whether the paper could be separated into two more focussed manuscripts, perhaps one focussing on the novel aspects of the post-processing method and validating its assumptions, and a second on evaluation the benefits and investigating factors that influence its performance."

We thank the reviewer for this suggestion. Although we have considered separating the paper into two papers, we prefer to keep the manuscript as one paper, as we feel it is important that the methods are discussed alongside their practical performance. However, we have shortened the paper (see response to RC1, particularly comments 5, 6, and 24).

More specific comments:

2. "The sample covariance matrix is used to characterise the joint distribution of the historic observations and water balance simulations, equation 7. There are potential issues that may be encountered using this approach and it would be good understand whether special treatments have been needed to overcome these."

   i. "The covariance matrix is computed over a set of historic observations and is likely to have inflated, or spurious, correlations over long lags if the seasonal cycle of streamflow is not considered."

We have added "Since many stations have short time-series the impact of the seasonal cycle on the joint distribution is not considered. Additionally, any spurious correlations resulting from these short time-series are not currently treated." to Section 3.3.2.

ii. "For large sample covariance matrices such as those estimated in this study, missing observations can lead covariance matrices that are not positive definite."

We have added "To ensure that the covariance matrix, $\Sigma_{\psi\psi}(k - q + 1 : k + T, k - q + 1 : k + T)$ is positive definite the minimum eigenvalue method is used (Tabeart et al., 2020) The covariance matrix is decomposed into the eigenvalues and eigenvectors. A minimum eigenvalue threshold is set as $1 \times 10^{-7}\lambda_1$ where $\lambda_1$ is the largest eigenvalue. All eigenvalues below this threshold are set to the threshold. The matrix is then reconstructed and scaled to match the variance of the original covariance matrix." to Section 3.3.2.

3. "The KGE analysis is performed using the median as a point estimate of the forecast ensemble. The results obtained for the post-processed forecasts, particularly the bias ratios and variability ratios of less than one at long lead times, are not unexpected as the variance of the forecast median will be considerably more damped that the mean. The forecast mean is likely to be a better choice as the point estimate of the forecast ensemble. Some theoretical justification of the use of the ensemble mean with measures of squared error can be found in Gneiting (2011)."

We thank the reviewer for this comment. We have now performed the calculations of the modified KGE for the ensemble mean and found that the distribution of the KGESS are similar for both point estimates. Therefore, we have kept the main analysis using the ensemble median. However, we have motivated the choice in Section 4.3.1: "The ensemble median of the raw forecasts is used in this evaluation because operationally the ensemble forecasts are often represented by boxplots where the median at each timestep is shown."

We found that the bias ratio for the ensemble mean is higher than those of the ensemble median and the post-processed forecast. Therefore, we have included the analysis of the components of the modified KGE for the ensemble mean in the supplementary material and added the following to section 5.1.1: "The ensemble mean is another commonly used single-valued summary of an ensemble forecast (Gneiting, 2011). Although the comparison presented here uses the ensemble median, we also show the three components of the *KGE'* for the ensemble mean in Fig. 1 of the supplementary material. The ensemble means (see Fig. 1b of the supplementary material) do not show the general drift in β-values with increasing lead-time that is discussed above for both the ensemble median and post-processed forecasts. However, the range of β-values is similarly large for both the ensemble median and the ensemble mean. In terms of the correlation coefficient and the variability ratio the ensemble mean performs similarly or worse than the ensemble median (see Fig. 1a and 1c of the supplementary material, respectively)."

4. "In this paper, the analysis of peak timing is conditioned on observations exceeding a threshold (90th percentile discharge threshold) within the forecast period and is likely to result in a biased evaluation of forecasts. A more rigorous approach would be to select the events based on forecasts exceeding the threshold."

We thank the reviewer for highlighting this limitation. We have run the analysis again using a forecast exceedance threshold and note that the bimodal distribution that was shown for lead-times 10-15 days is no longer present. We believe the bimodal nature was due to forecasts that failed to predict an event being more harshly penalised than forecasts that predicted an event that did not occur. We have changed the peak-time error analysis to use a forecast exceedance threshold.

5. "I also believe that rather than evaluating the timing of the peak in the forecast median, which doesn't correspond to the peak in any individual hydrograph, a more representative point

estimate of the forecast timing error would be to compare the median (or mean) time to peak across all ensemble members to the timing of the observed peak. "

We thank the reviewer for this comment. We have performed the analysis with both the peak of the ensemble median and the median of the peaks of each of the ensemble members using the forecast-based threshold (see response to comment 4). We found that the distributions are very similar with the ensemble median performing slightly better at longer lead-times. Therefore, we have kept the main comparison between the post-processed forecast and the ensemble median, but we have briefly mentioned the median of the peaks for all ensemble members as an alternative.

6. "line 373 - values in the recent perion should be "values in recent period"
We thank the reviewer for highlighting this mistake and have corrected this mistake.

7. "Line 825 - CRPS calculated on deterministic forecasts is equivalent to the absolute error not the square absolute error."
We changed "square of the absolute difference" to the "absolute error" on line 797.

8. "Figures - The size of multi-panel figures (e.g. Figure 9, 12) could be increased to better illustrate the detail"
We thank the reviewer for highlighting the figures which are unclear. We have increased the size of Figures 6, 9, and 12.

References:

Zamo, M., Naveau, P. Estimation of the Continuous Ranked Probability Score with Limited Information and Applications to Ensemble Weather Forecasts. Math Geosci **50,** 209–234 (2018). https://doi.org/10.1007/s11004-017-9709-7)

Ferro, C.A.T., Richardson, D.S. and Weigel, A.P. (2008), On the effect of ensemble size on the discrete and continuous ranked probability scores. Met. Apps, 15: 19-24. https://doi.org/10.1002/met.45

Skøien, J. O., Bogner, K., Salamon, P., & Wetterhall, F. (2021). On the Implementation of Postprocessing of Runoff Forecast Ensembles, Journal of Hydrometeorology, 22(10), 2731-2749

Mazzetti, C., Decremer, D., Prudhomme, C. Challenges of the European Flood Awareness System (EFAS) hydrological calibration. Poster presented at: Joint Virtual Workshop on "Connecting global to local hydrological modelling and forecasting: scientific advances and challenges"; June 29, 2021; Online [Available: https://events.ecmwf.int/event/222/overview].

MacDonald, A., Scarrott, C. J., Lee, D., Darlow, B., Reale, M., and Russell, G.: A flexible extreme value mixture model, Computational Statistics & Data Analysis, 55, 2137–2157, 2011.

Bogner, K., Pappenberger, F., and Cloke, H.: The normal quantile transformation and its application in a flood forecasting system, Hydrology and Earth System Sciences, 16, 1085–1094, 2012.

Coccia, G.: Analysis and developments of uncertainty processors for real time flood forecasting, Ph.D. thesis, alma, http://amsdottorato.unibo.it/3423/, 2011.